# Influence of patient characteristics on chimeric antigen receptor T cell therapy in B-cell acute lymphoblastic leukemia

Furun An [1,2,6], Huiping Wang [1,2,6], Zhenyun Liu [2,3], Fan Wu [1,2], Jiakui Zhang [1,2], Qianshan Tao[1,2], Yingwei Li [1,2], Yuanyuan Shen [1,2], Yanjie Ruan [1,2], Qing Zhang [1,2], Ying Pan[1,2], Weiwei Zhu [1,2], Hui Qin [1,2], Yansheng Wang [1,2], Yongling Fu[3], Zhenqing Feng [4,5✉] & Zhimin Zhai [1,2✉]

CD19-specific chimeric antigen receptor T cell (CD19 CAR T) therapy has shown high remission rates in patients with refractory/relapsed B-cell acute lymphoblastic leukemia (r/r B-ALL). However, the long-term outcome and the factors that influence the efficacy need further exploration. Here we report the outcome of 51 r/r B-ALL patients from a non-randomized, Phase II clinical trial (ClinicalTrials.gov number: NCT02735291). The primary outcome shows that the overall remission rate (complete remission with or without incomplete hematologic recovery) is 80.9%. The secondary outcome reveals that the overall survival (OS) and relapse-free survival (RFS) rates at 1 year are 53.0 and 45.0%, respectively. The incidence of grade 4 adverse reactions is 6.4%. The trial meets pre-specified endpoints. Further analysis shows that patients with extramedullary diseases (EMDs) other than central nervous system (CNS) involvement have the lowest remission rate (28.6%). The OS and RFS in patients with any subtype of EMDs, higher Tregs, or high-risk genetic factors are all significantly lower than that in their corresponding control cohorts. EMDs and higher Tregs are independent high-risk factors respectively for poor OS and RFS. Thus, these patient characteristics may hinder the efficacy of CAR T therapy.

[1] Hematology Department, the Second Hospital of Anhui Medical University (SHAMU), Hefei, Anhui Province, China. [2] Hematologic Diseases Research Center of Anhui Medical University, Hefei, Anhui Province, China. [3] Sinobioway Cell Therapy Co., Ltd., Hefei, Anhui Province, China. [4] Key Laboratory of Antibody Technique of National Health Commission, Nanjing Medical University, Nanjing, Jiangsu Province, China. [5] Jiangsu Key Lab. of Cancer Biomarkers, Prevention and Treatment, Collaborative Innovation Center for Cancer Personalized Medicine, Nanjing Medical University, Nanjing, Jiangsu Province, China. [6] These authors contributed equally: Furun An, Huiping Wang. ✉email: fengzhenqing@njmu.edu.cn; zzzm889@163.com

C D19-specific chimeric antigen receptor T-cell therapy (CD19 CAR T) has shown high rates of initial response among children and young adults with refractory/relapsed B-cell acute lymphoblastic leukemia (r/r B-ALL)[1–6]. However, growing experience with these agents has revealed that remissions will be brief in a substantial number of patients[7–13]. Up to now, the factors for precluding durable remissions following CAR T-cell therapy mainly focus on poor CAR T-cell persistence and/or tumor cell resistance due to antigen loss or modulation, patient's related factors are rarely considered or reported[13]. Exploration of patient characteristics that may influence or predict outcomes could be helpful to understand objectively the limitations of this novel therapy. Besides, it can enable the development and manufacture of patient-specific optimized therapies. From November 2015 to May 2019, we conducted a phase II clinical trial (#NCT02735291) involving pediatrics and adults with r/r B-ALL who were treated with CD19 CAR T (termed Sino19 cell). In this work, the overall remission rate is 80.9%; the overall survival (OS) and relapse-free survival (RFS) rates at 1 year are 53.0 and 45.0% (median OS and RFS are 415.0 and 319.0 days), respectively. Patients with extramedullary diseases (EMDs) other than the central nervous system (CNS) involvement have the lowest remission rate (28.6%) comparing with patients with isolated CNS involvement and those without EMDs. The OS and RFS in patients with any subtype of EMDs, higher regulatory T cells (Tregs), or high-risk genetic factors are lower. Multivariable analyses reveal that EMDs and higher Tregs are independent high-risk factors respectively for poor OS and RFS, so these patient factors may preclude the efficacy of CAR T therapy in r/r B-ALL.

## Results

**Patients.** From November 2015 to May 2019, a total of 72 patients with r/r B-ALL were screened, 51 who met the inclusion criteria were enrolled; 4 patients did not have Sino 19 cell infusion due to production failure. Ultimately, 47 patients received Sino 19 cell infusion and were followed up for at least 1 month for efficacy and safety assessment (Fig. 1). The age of these 47 patients ranges from 3 to 72 and the median age was 22, including 23 males and 24 females. 3 (3/47, 6.4%) had primary refractory B-ALL; 9 (9/47, 19.1%) had previously received allogeneic hematopoietic stem cell transplantation (allo-HSCT) and 3 of them used blinatumomab also; 28 (28/47, 59.6%) had cytogenetic and/or molecular aberrations predicting poor prognosis; 13 (13/47, 27.7%) had active EMDs at enrollment. In total, 45 patients received preconditioning chemotherapy for lymphodepletion and to lower tumor burden; 2 elderly patients (aged 66 and 72 years old) were exempted (Table 1).

**Primary outcomes.** The overall remission rate was 80.9% (38/47) within 2 months after the infusion of Sino 19 cells (Table 2). Among the 38 patients who acquired remission which includes complete remission (CR) or CR with incomplete hematologic recovery (CRi), 1 was positive for minimal residual disease (MRD) (0.29%), and this patient relapsed 180 days post infusion. A few patients who relapsed and/or became *BCR/ABL* fusion gene positive after remission received mild salvage chemotherapy and/or tyrosine kinase inhibitor treatment if the Sino 19 cells in vivo had been lost.

**Secondary outcomes.** The estimated median OS and RFS were 415.0 and 319.0 days, the rates of OS and RFS at 1 year were 53.0 and 45.0%, respectively (Figs. 2a and 3a). There was no significant difference between the pediatric and adult cohort with regards to

CR/CRi rate (76.5% vs. 83.3%, $p = 0.850$), OS ($p = 0.329$), or RFS ($p = 0.168$) (Figs. 2b and 3b).

**Safety.** The incidence of grade 4 adverse reactions was 6.4% (3/47) (one patient with grade 4 immune effector cell-associated neurotoxicity syndrome (ICANS) and two patients with grade 4 ALT/AST increases). Eleven patients (23.4%) had grade 3 cytokine release syndrome (CRS); serum IL-6 levels were significantly elevated in 3 of them post-infusion (greater than 100 times of baseline values), and were treated with IL-6R antagonist tocilizumab. The remaining eight patients had either normal or slightly increased serum IL-6 levels; they were managed with corticosteroids and supportive care.

Only 2 patients (4.3%) had ICANS, one patient with complicated CNS involvement only with no movement disorder before Sino 19 cell infusion, developed grade 3 CRS after the infusion, this patient had a fever, hypotension requiring a vasopressor, etc. and recovered 7 days after symptomatic treatment. On the 31st day, post-infusion achieved CR in bone marrow (BM) and cerebrospinal fluid (CSF) (blast cells and MRD were all negative), but on the 34th day, this patient began feeling weakness and numbness from the lower limbs, then gradually progressed paralysis of both lower extremities. Ten days later the paralysis and numbness stabled at about the eighth thoracic level. On the 180th day, over 10% blasts were detected in blood and BM

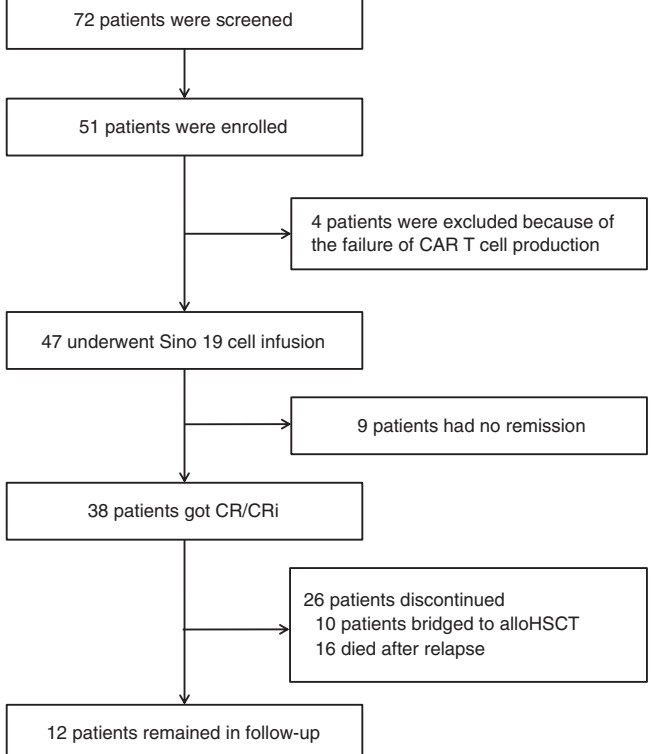

**Fig. 1 Consort diagram.** A total of 72 patients with r/r B-ALL were screened, 51 who met the inclusion criteria and did not meet the exclusion criteria were enrolled; 4 patients did not have Sino 19 cell infusion due to production failure. Ultimately, 47 patients received Sino 19 cell infusion; 9 of them had no remission. 38 patients achieved CR/CRi; 10 of them were bridged to allo-HSCT; 19 out of the remaining 28 patients who were not bridged to allo-HSCT relapsed, out of which 16 died. Twelve patients remained in follow-up at the time of data cut off, which included 3 relapsed patients and 9 with sustained remission. Source data is provided as a Source Data file or available at https://doi.org/10.6084/m9.figshare.13136078.v1.

**Table 1 Characteristics of evaluable patients[a].**

| Characteristics | Pediatric (n = 17) | Adult (n = 30) | Total (n = 47) |
|---|---|---|---|
| *Sex—no. (%)* | | | |
| Female | 8 (47.1) | 16 (53.3) | 24 (51.1) |
| Male | 9 (52.9) | 14 (46.7) | 23 (48.9) |
| *Age at infusion—y* | | | |
| Median | 7.0 | 32.0 | 22.0 |
| Range | 3–13 | 14–72 | 3–72 |
| *Previous chemotherapy—no. (%)* | | | |
| ≥10 | 8 (47.1) | 11 (36.7) | 19 (40.4) |
| <10 | 9 (52.9) | 19 (63.3) | 28 (59.6) |
| *Previous allo-HSCT—no. (%)* | | | |
| Yes | 4 (23.5) | 5 (16.7) | 9 (19.1) |
| No | 13 (6.5) | 25 (83.3) | 38 (80.9) |
| Primary refractory disease—no. (%) | 0 | 3 (10.0) | 3 (6.4) |
| *Relapse—no. (%)* | | | |
| 1 time | 12 (70.6) | 13 (43.3) | 25 (53.2) |
| ≥2 times | 5 (29.4) | 14 (46.7) | 19 (40.4) |
| *Percentage of blasts in bone marrow—no. (%)* | | | |
| <5% | 4 (23.5) | 11 (36.7) | 15 (31.9) |
| ≥5% and <20% | 2 (11.8) | 4 (13.3) | 6 (12.8) |
| ≥20% | 11 (64.7) | 15 (50.0) | 26 (55.3) |
| *Cytogenetic factors[b]—no. (%)* | | | |
| No data | 1 (5.9) | 2 (6.7) | 3 (6.4) |
| Undetermined significance | 2 (11.8) | 3 (10.0) | 5 (10.6) |
| *TEL/AML1* | 1 (5.9) | 0 | 1 (2.1) |
| Normal | 4 (23.5) | 6 (20.0) | 10 (21.3) |
| Ph+ or *BCR/ABL1* with T315I mutation | 0 | 4 (13.3) | 4 (8.5) |
| Ph+ or *BCR/ABL1* without T315I mutation | 2 (11.8) | 5 (16.7) | 7 (14.9) |
| *BCR/ABL like* fusions | 2 (11.8) | 1 (3.3) | 3 (6.4) |
| *MLL* translocation | 1 (5.9) | 2 (6.7) | 3 (6.4) |
| *E2A/PBX1* | 1 (5.9) | 4 (13.3) | 5 (10.6) |
| *HOX11* | 0 | 1 (3.3) | 1 (2.1) |
| Complex karyotype | 3 (17.6) | 2 (6.7) | 5 (10.6) |
| Extramedullary diseases (EMDs) before infusion[c]—no. (%) | 3 (17.6) | 10 (33.3) | 13 (27.7) |
| CNS involvement (central nervous system involvement) only | 0 | 6 (20.0) | 6 (12.8) |
| Testicle | 1 (5.9) | 1 (3.3) | 2 (4.3) |
| Breast | 0 | 1 (3.3) | 1 (2.1) |
| CNS involvement and multi-site bone infiltration simultaneously | 0 | 1 (3.3) | 1 (2.1) |
| Multi-site bone and soft-tissue infiltration simultaneously | 2 (11.8) | 1 (3.3) | 3 (6.4) |
| Percentage of regulatory T cells in peripheral blood[d]—no. (%) | | | |
| ≥Upper limits of normal value | 10 (58.8) | 18 (62.1) | 28 (60.9) |
| <Upper limits of normal value | 7 (41.2) | 11 (37.9) | 18 (39.1) |
| *Preconditioning chemotherapy[e]—no. (%)* | | | |
| FC | 9 (52.9) | 16 (53.3) | 25 (53.2) |
| VDCP | 1 (5.9) | 5 (16.7) | 6 (12.8) |
| Cy | 7 (41.2) | 7 (23.3) | 14 (29.8) |
| Not given | 0 | 2 (6.7) | 2 (4.3) |

[a]Percentages may not total 100 because of rounding.
[b]Ph+ means Philadelphia chromosome positive.
[c]Extramedullary diseases (EMDs) were defined as CD19+ B lymphoblastic cells outside the bone marrow, detected by revealing abnormal masses in extramedullary sites. Imaging examinations, such as magnetic resonance imaging, computed tomography or positron-emission tomography, and cytological or pathological biopsy examinations were used to detect blast cells in these masses (Fig. 6). CNS involvement (central nervous system involvement) was defined as blast cells detected in cerebrospinal fluid (CSF sample with ≥5 leukocytes per cubic millimeter and <10 erythrocytes per cubic millimeter).
[d]Tregs data were not available for one patient.
[e]FC (fludarabine 25 mg m$^{-2}$ followed by cyclophosphamide 400 mg m$^{-2}$ × 5 days) was used if a patient had normal white blood cell count and active bone marrow hyperplasia; VDCP (vincristine 1 mg m$^{-2}$ + daunorubicin 60 mg m$^{-2}$ + cyclophosphamide 600 mg m$^{-2}$) × 1 day + prednisone 2 mg kg$^{-1}$ × 7 days then followed by FC × 3 days, if a patient had higher tumor burden such as percentage of marrow blasts ≥50% or EMD positive; Cy (cyclophosphamide, 400 mg m$^{-2}$ × 3 days), if a patient had leukopenia and hypoplastic marrow; not given, for two aged patients.
Source data are provided as a Source Data file or available at https://doi.org/10.6084/m9.figshare.13136078.v1.

samples; on the 209th day, this patient died of leukemia progression. The patient showed signs of neurological damage, but repeated imaging examinations found no occupying lesions or other abnormalities, had no evidence of CNS infection by microbiological examinations, and CSF monthly examinations did not reveal any abnormal cells. Lastly, the Guillain–Barre syndrome was also excluded because there was no history of prodromal infection and protein–cell separation phenomenon, etc., so we think the spinal cord injury was related to CAR T toxicity and made a diagnosis of ICANS grade 4 (details about this patient can be taken from the "Response to Referees" in Supplementary files). Some patients had transient increases in aminotransferase and/or creatinine levels; abnormal coagulation time, other abnormal metabolic testings, etc. (Table 3). Two of them reached 4-grade transaminase increase. The nine patients who had previously undergone allo-HSCT did not develop graft-versus-host disease after the Sino 19 cell infusion.

To monitor the development and duration of B-cell aplasia, we detected CD45-strong positive and CD19-positive mature B cells by flow cytometry (Fig. 4b). B-cell aplasia occurred in all the patients who had CR/CRi and persisted from 44 days to 423 days post infusion. Despite regular immunoglobulin replacement, 24 B-cell aplasia patients developed various infections within 6 months after the infusion: 39% (15/38) had bronchitis or pneumonia, 11% (4/38) had cystitis, and 13% (5/38) had other infections such as herpes zoster and tympanitis. All infections were appropriately controlled with proper and prompt treatment.

**Results of exploration.** To explore the relevant factors affecting the efficacy, we routinely analyzed whether the CR, OS, and RFS were related to patient's baseline characteristics such as gender, age, number of cycles of chemotherapy, relapse, previous allo-HSCT, and proportion of BM blast cells; all results did not show significant differences (Table 2; Figs. 2b and 3b). Then we further ad-hoc analyzed the other potential influence factors on efficacy, children and adults who were taken as a whole cohort.

*EMDs*: A total of 13 patients had active EMDs before the infusion. Among them, six patients had EMD confined to just CNS involvement, and seven had EMDs other than CNS that included one patient with CNS involvement and a multi-site bone infiltration simultaneously (Table 1). Patients without EMDs showed a better CR/CRi rate than patients with EMDs (91.2% vs. 53.8%, $p = 0.013$). Furthermore, the CR/CRi rate in patients with EMDs other than CNS was the lowest (28.6%); no significant difference existed between patients with CNS involvement only and the patients without EMDs (83.3% vs. 91.2%, $p = 0.493$) (Table 2). About survival, the OS and RFS in patients without EMDs were higher than that in patients with CNS involvement only ($p = 0.021$, $p = 0.003$) or in patients with EMDs other than CNS ($p = 0.015$, $p = 0.003$), respectively, there was no significant difference between the latter two subgroups ($p = 0.905$ and $p = 0.961$, respectively) (Figs. 2c and 3c).

*Abnormally increased levels of Tregs*: Out of the 47 evaluable patients, circulating Tregs analysis by flow cytometry was performed for 46 after conditioning chemotherapy and before Sino 19 cell infusion. Twenty-eight (60.9%) of the patients had abnormally increased circulating Tregs, and the other 18 had either normal or lower Tregs levels. The CR/CRi rate in patients with higher circulating Tregs was lower than in patients with normal or lower Tregs, but it was not statistically significant (75.0% vs. 88.9%, $p = 0.191$, Table 2). The OS and RFS rates at 1 year in patients with higher Tregs were significantly lower compared to that in patients with normal or lower Tregs (29.3%

**Table 2 Comparison of remission rate according to patient's clinical characteristics.**

| | Patients no. | Remission no. (%) | 95% CI | $\chi^2$ | p Value |
|---|---|---|---|---|---|
| Overall | 47 | 38 (80.9) | 69.2–92.5 | | |
| *Sex* | | | | 2.415 | 0.120 |
| M | 23 | 16 (69.6) | 49.2–89.9 | | |
| F | 24 | 22 (91.7) | 79.7–103.6 | | |
| *Age* | | | | 0.036 | 0.850 |
| Pediatric cohort (<14 y) | 17 | 13 (76.5) | 54.0–99.0 | | |
| Adult cohort (≥14 y) | 30 | 25 (83.3) | 69.2–97.5 | | |
| *Previous allo-HSCT* | | | | 0.044 | 0.833 |
| Yes | 9 | 8 (88.9) | 63.3–114.5 | | |
| No | 38 | 30 (78.9) | 65.4–92.5 | | |
| *Times of previous chemotherapy* | | | | 0.424 | 0.515 |
| ≥10 | 19 | 14 (73.7) | 51.9–95.5 | | |
| <10 | 28 | 24 (85.7) | 71.9–99.5 | | |
| *Refractory* | | | | – | 1.000[a] |
| Yes | 3 | 3 (100.0) | 100.0 | | |
| No | 44 | 35 (79.5) | 67.1–92.0 | | |
| *Times of relapse* | | | | 0.214 | 0.643 |
| 1 | 25 | 21 (84.0) | 68.6–99.4 | | |
| ≥2 | 19 | 14 (73.7) | 51.9–95.5 | | |
| *Percentage of marrow blasts ≥ 5%* | | | | 0 | 1.000 |
| Yes | 32 | 26 (81.3) | 67.0–95.5 | | |
| No | 15 | 12 (80.0) | 57.1–102.9 | | |
| *Percentage of marrow blasts ≥ 20%* | | | | 0.151 | 0.698 |
| Yes | 26 | 20 (76.9) | 59.6–94.3 | | |
| No | 21 | 18 (85.7) | 69.4–102.0 | | |
| *High-risk cytogenetic factors*[b] | | | | – | 0.655[a] |
| Yes | 28 | 23 (82.1) | 67.0–97.3 | | |
| No | 11 | 10 (90.9) | 70.7–111.2 | | |
| *Active EMDs before infusion* | | | | 6.225 | 0.013[c] |
| Yes | 13 | 7 (53.8) | 22.5–85.2 | | |
| No | 34 | 31 (91.2) | 81.1–101.2 | | |
| *Subgroup 1: CNS involvement only vs. no-EMDs* | | | | – | 0.493[a] |
| CNS involvement | 6 | 5 (83.3) | 40.5–126.2 | | |
| No-EMDs | 34 | 31 (91.2) | 81.1–101.2 | | |
| *Subgroup 2: CNS involvement only vs. EMDs other than CNS*[d] | | | | – | 0.103[a] |
| CNS involvement only | 6 | 5 (83.3) | 40.5–126.2 | | |
| EMDs other than CNS | 7 | 2 (28.6) | −16.6 to 73.7 | | |
| *Subgroup 3: EMDs other than CNS vs. no-EMDs* | | | | 10.775 | 0.001[c] |
| EMDs other than CNS | 7 | 2 (28.6) | −16.6 to 73.7 | | |
| No-EMDs | 34 | 31 (91.2) | 81.1–101.2 | | |
| *Tregs before infusion*[e] | | | | 0.605 | 0.437 |
| Higher cohort | 28 | 21 (75.0) | 57.9–92.1 | | |
| Normal or lower cohort | 18 | 16 (88.9) | 72.8–105.0 | | |

[a]Fisher's exact test
[b]Three patients did not have data on genetic factors, and five patients had cytogenetic/molecular abnormalities of undetermined significance.
[c]Compared with patients without EMDs, the CR/CRi rate in patients with EMDs was significantly lower (91.2% vs. 53.8%, p = 0.013) with a statistical power of 82.9%, the CR/CRi rate in patients with EMDs other than CNS was the lowest (91.2% vs. 28.6%, p = 0.001) with a statistical power of 96.8%, but the CR/CRi rate in patients with CNS involvement only had no significant difference (91.2% vs. 83.3%, p = 0.493) with a statistical power of 9.1%.
[d]The locations of these EMDs other than CNS, include three patients with bone and soft tissue multiple locally involvement, two with testicular involvement, one in the right breast, and one in local bone concurrently with CNS involvement.
[e]Tregs data were not available for one patient.
A two-sided Pearson chi-squared test (or Fisher's exact test) was used for comparison between groups. Source data is provided as a Source Data file or available at https://doi.org/10.6084/m9.figshare.13136078.v1.

vs. 64.2%, p = 0.011, Fig. 2d; 11.9% vs. 56.7%, p = 0.002, Fig. 3d), respectively. In addition, the Tregs level was associated with EMDs (Fig. 5), but the lymphodepletion regimen did not influence the Tregs level (Supplementary Tables 1 and 2). Patients without EMDs had lower Tregs compared to patients with CNS involvement only (p = 0.011) or EMDs other than CNS (p = 6.42e−5); however, there was no significant difference between the two EMDs subgroups (p = 0.237). Other factors may influence the circulating Tregs level after infusions such as the composition of Sino 19 cells, treatment of CRS/ICANS, and baseline Tregs level. These factors will be further elaborated on in our future reports.

*Cytogenetic and molecular abnormalities*: Overall, 28 patients were found to have one or more significantly high-risk cytogenetic/molecular abnormalities (Table 1). Their CR/CRi rate did not show a significant difference compared with the patients without high-risk cytogenetic/molecular abnormalities (82.1% vs. 90.9, p = 0.850, Table 2); but the OS and RFS rates at 1 year were significantly lower (34.3% vs. 66.7%, p = 0.047, Fig. 2e; 24.2% vs. 67.7%, p = 0.034, Fig. 3e), respectively.

*Allo-HSCT after CAR T cell therapy*: Ten patients who achieved CR/CRi after CAR T subsequently received allo-HSCT; we analyzed the role of the allo-HSCT on survival using the time-dependent Cox model. The hazard ratio (HR) of allo-HSCT on

**Table 3 Non-hematological adverse events within 4 weeks[a].**

|  | Total (%) | Grade 3–4(%) |
|---|---|---|
| CRS | 39 (83.0) | 11 (23.4) |
| ICANS | 2 (4.3) | 1 (2.1) |
| ALT/AST increased | 10 (21.3) | 2 (4.3) |
| CRE increased | 2 (4.3) | 0 |
| APTT/PT prolonged | 10 (21.3) | 0 |
| Fibrinogen decreased | 5 (10.6) | 0 |
| Acidosis | 1 (2.1) | 0 |
| Hypokalemia | 4 (8.5) | 1 (2.1) |
| Hypoglycemia | 2 (4.3) | 0 |
| Cephalalgia | 2 (4.3) | 0 |
| Myalgia | 2 (4.3) | 0 |

[a]According to the CTCAE grading system by the United States National Cancer Institute[38], and ASBMT Consensus Grading for Cytokine Release Syndrome and Neurologic Toxicity Associated with Immune Effector Cells[39].
*CRS* represents cytokine release syndrome, *ICANS* immune effector cell-associated neurotoxicity syndrome, *ALT* alanine aminotransferase, *AST* aspartate aminotransferase, *CRE* creatinine, *APTT* activated partial thromboplastin time, *PT* prothrombin time.
Source data is provided as a Source Data file or available at https://doi.org/10.6084/m9.figshare.13136078.v1.

OS and RFS was 0.187 (95% CI: 0.025–1.420) and 0.533 (95% CI: 0.168–1.689), respectively, indicating that allo-HSCT might be a protective factor for OS and RFS, though not statistically significant ($p = 0.105$ for OS and $p = 0.285$ for RFS).

*Persistence of Sino 19 cell*: Real-time quantitative reverse transcription PCR (qRT-PCR) was regularly used to analyze the CAR DNA of Sino 19 cell from whole blood or CSF samples (Fig. 4a). The median persistence time of Sino 19 cells for all patients that attained CR/CRi was 85 days (range: 44–498 days) excluding 10 patients who were bridged to allo-HSCT. The persistence time correlated well with the duration of B cell aplasia ($p = 1.08e-7$, $R = 0.884$) (Fig. 4c). This indicates that B cell loss and recovery could be used as a pharmacodynamic measure of CAR T function.

In total, 19 patients relapsed; 78.9% (15/19) happened after Sino 19 cell loss; the other 4 (21.1%) relapsed while Sino 19 cell was persisting in vivo, which included one CD19 negative relapse. These four patients all had EMDs, abnormally increased Tregs, and high-risk genetic abnormalities simultaneously. For patients who maintained their remission up to the data cut-off time, 66.7% (6/9) lost Sino 19 cells. Interestingly, Sino 19 cells and B-cell

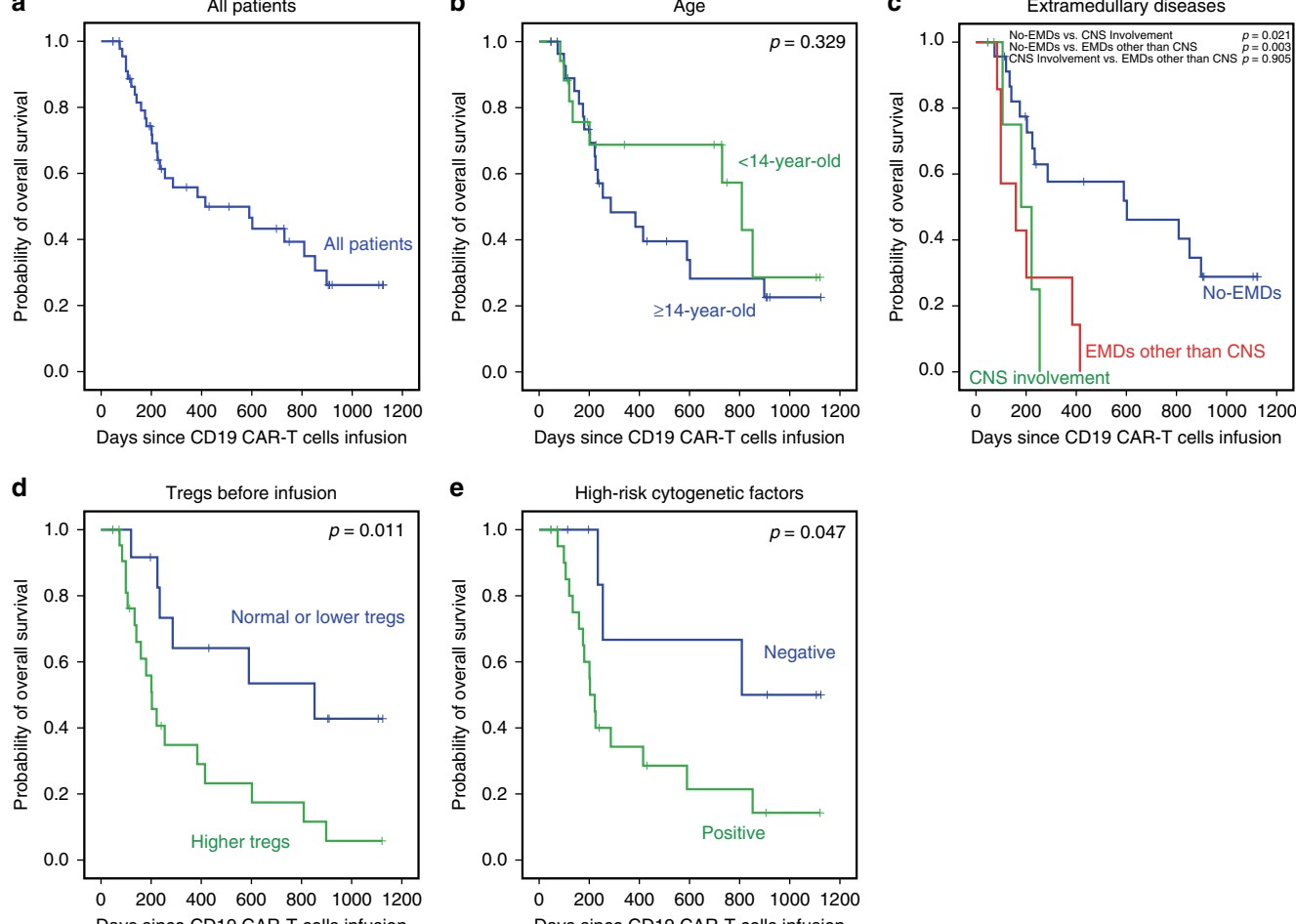

**Fig. 2 Overall survival (OS) and subgroup analysis. a** The OS in all patients who received Sino 19 infusion, the median OS was 415.0 days (95% CI: 15.5–814.5). The rate of OS at 1 year was 53.0% (95% CI: 37.2–68.8). **b** The OS between pediatric and adult cohort showed no significant difference ($p = 0.329$). **c** The OS in patients with EMDs (CNS involvement only or EMDs other than CNS) was lower than that in patients without EMDs ($p = 0.021$ and 0.003), respectively, but no significant difference was observed between the two EMDs subgroups ($p = 0.905$). **d** The OS in patients with or without higher Tregs, 1-year OS rate was 29.3% vs. 64.2% ($p = 0.011$). **e** The OS in patients with or without high-risk cytogenetic factors, 1-year OS rate was 34.3% vs. 66.7% ($p = 0.047$). Patients who were bridged to allo-HSCT were excluded when we analyzed the influence of patient characteristics on OS (**c–e**). **b–e** were performed by the Kaplan–Meier approach and used a two-sided log-rank test. Source data is provided as a Source Data file or available at https://doi.org/10.6084/m9.figshare.13136078.v1.

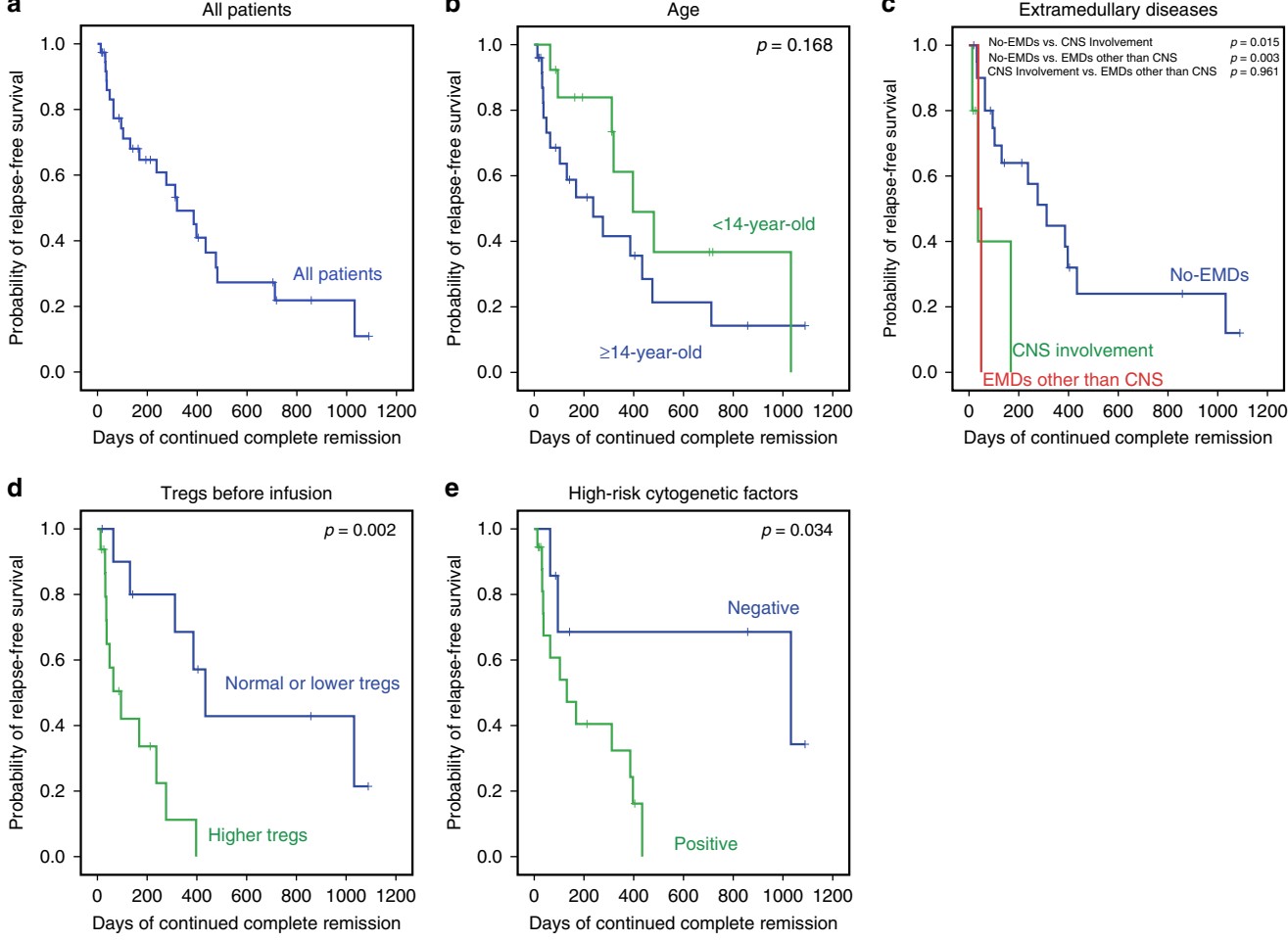

**Fig. 3 Relapse-free survival (RFS) and subgroup analysis. a** The RFS in all patients who received infusion, the median RFS was 319.0 days (95% CI: 182.6–455.4). The rate of RFS at 1 year was 45.0% (95% CI: 26.8–63.2). **b** The RFS between pediatric and adult cohort, no significant difference observed (*p* = 0.168). **c** The RFS in patients with the 2 subtypes of EMDs (CNS involvement only or EMDs other than CNS) were lower than that in patients without EMDs (*p* = 0.015 and 0.003), respectively, but no significant difference existed between the subtypes (*p* = 0.961). **d** The RFS in patients with or without higher Tregs before infusion, 1-year RFS rate was 11.9% vs. 56.7% (*p* = 0.002). **e** The RFS in patients with or without high-risk cytogenetic factors, 1-year RFS rate was 24.2% vs. 67.7% (*p* = 0.034). Patients who were bridged to allo-HSCT were excluded when we analyzed the influence of patient characteristics on RFS (**c–e**). **b–e** were performed by the Kaplan–Meier approach and used a two-sided log-rank test. Source data is provided as a Source Data file or available at https://doi.org/10.6084/m9.figshare.13136078.v1.

aplasia were detected in 9 unresponsive/refractory patients within 60 days post infusion. Most of them (88.9%, 8/9) had either EMDs (1 with CNS involvement only, 5 with multi lesions other than CNS) or high-risk cytogenetic/molecular abnormalities (2 positives for *E2A/PBX1*). The 9th patient had no EMDs, but the cytogenetic /molecular data for this patient was not available. Sino 19 cells were successfully detected in only 7 patients CSF samples, which included 3 patients with CNS involvement from day 28 to day 180 after the infusion (generally, nucleated cells in CSF is lower), all of them achieved remission. Due to the small number of patients and limited observation time, we could not further analyze the relationship between Sino 19 cell persistence or B-cell aplasia and patient outcomes.

*Multivariable analyses*: EMDs, higher Tregs, and high-risk cytogenetic factors were significantly associated with survival by Kaplan–Meier analysis. In order to determine if these factors had an independent effect on survival after CAR T unitary therapy, we first excluded patients who were bridged to allo-HSCT, then introduced the three variables to the Cox proportional hazard model for analysis. The results show that EMDs was an independent poor factor for OS (*p* = 0.004), patients with EMDs

had a higher risk of death (HR = 4.551, 95% CI: 1.605–12.907), with a statistical power of 86.2%; higher Tregs was an independent poor factor for RFS (*p* = 0.010), patients with higher Tregs had a higher risk of relapse (HR = 5.096, 95% CI: 1.479–17.558), with a statistical power of 93.2%.

## Discussion

Historically traditional chemotherapy has unsatisfactory treatment outcomes for r/r B-ALL patients, especially for those who relapse after HSCT[14,15]. Multiple clinical trials have reported remarkable CR rates ranging from 70 to 90% using CD19-specific CAR T cells[1–12,16]. Despite these promising clinical outcomes, a subset of patients relapses and/or develops unique toxicities such as severe CRS and ICANS on this potentially curative therapy, thus becoming another focus of increasing concern[17,18].

On safety, the total incidence of grade 4 adverse reactions was 6.4% in our trial; 23.4% of the patients had grades 3 CRS after Sino 19 cell infusion. All patients with severe CRS were successfully treated with tocilizumab and/or dexamethasone. The incidence of ICANS was 4.3%; one patient expressed infrequent

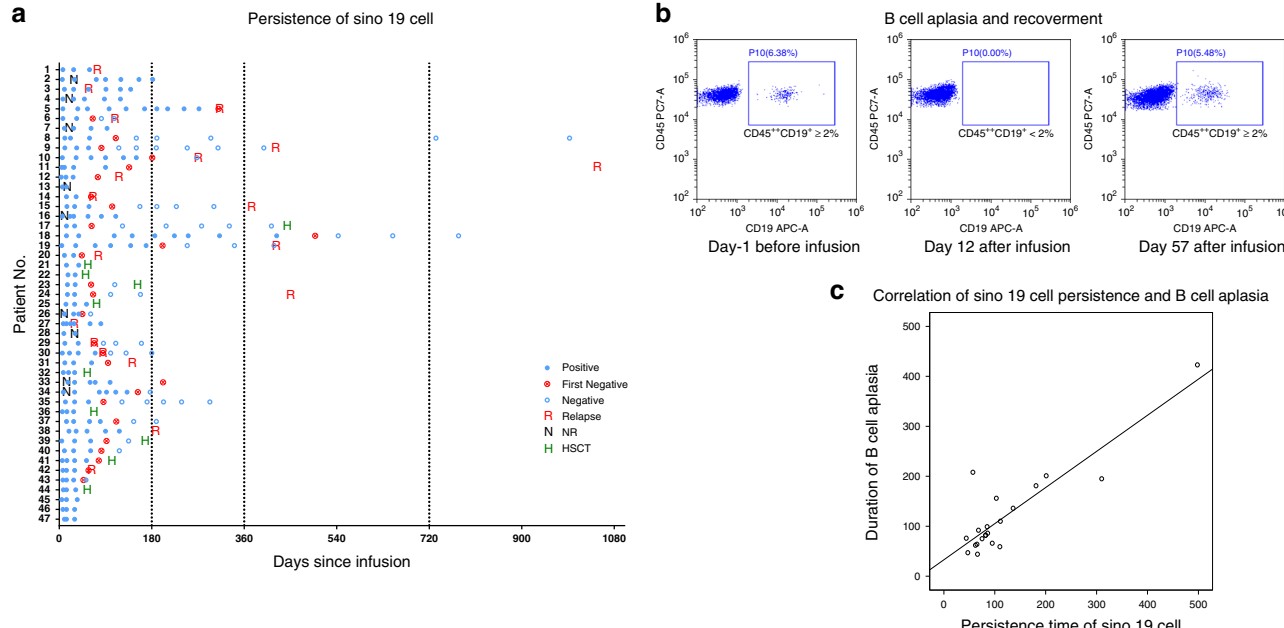

**Fig. 4 Persistence of Sino 19 cell and B-cell aplasia. a** Panel shows the results of Sino 19 cells detected by qRT-PCR in peripheral blood samples. The first negative was defined as the time of first negative measurement by qRT-PCR. The median persistence time of Sino 19 cells for all patients who attained CR/CRi was 85 days (range 44–498 days), excluding 10 patients that were bridged to allo-HSCT. 15 (78.9%) patients relapsed after the Sino 19 cell loss or at the same time; another 4 (21.1%) relapsed under the state of Sino 19 cell persistence (Nos. 1, 3, 27, and 38). Nine patients (Nos. 2, 4, 7, 13, 16, 26, 28, 33, and 34) failed to achieve CR/CRi (indicated by NR), however, Sino 19 cell was detected in their blood from day 1 to day 60 after the infusion. Two patients (Nos. 8 and 18) who did not bridge to allo-HSCT maintained continue remission and survived for more than 1 year after the Sino 19 cell loss. **b** The panel shows the detection of B cell in a patient before and after the infusion of Sino 19 cells; B-cell aplasia was defined as CD45 strong and CD19-positive (CD19$^+$ CD45$^{++}$) B cells <2% in lymphocyte gate; recovery was defined as ≥2%. **c** The panel shows that the persistence time of Sino 19 cells positively well correlated with the duration of B cell aplasia in patients who achieved CR/CRi without bridging to allo-HSCT ($p = 1.08e-7$, $R = 0.884$, by using two-sided Pearson correlation coefficient). Source data of (**a**) and (**c**) are provided as a Source Data file or available at https://doi.org/10.6084/m9. figshare.13136078.v1.

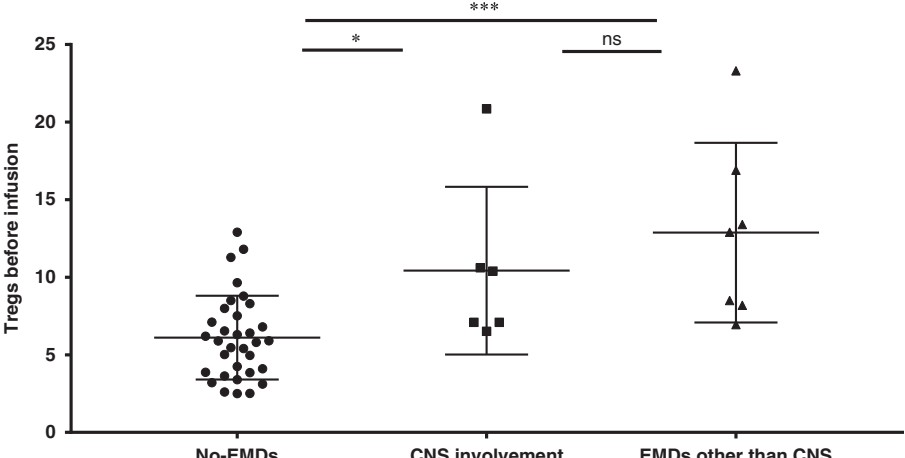

**Fig. 5 Correlations between Tregs and EMDs.** The circulating Tregs in patients with CNS involvement (10.43 ± 5.41, $n = 6$) or EMDs other than CNS (12.88 ± 5.79, $n = 7$) all higher than that in patients without EMDs (6.11 ± 2.70, $n = 33$) ($p = 0.011$ and $6.42e-5$) respectively, but no significant difference observed between the two subtypes ($p = 0.237$). Tregs levels were all measured by flow cytometry during post-conditioning chemotherapy and before Sino 19 cells infusion. Error bars represent mean ± SD; * represents $p < 0.05$; *** represents $p < 0.001$; ns represents no significance; mean between groups were compared by one-way ANOVA and multiple comparisons were compared using LSD tests (two-sided). $p < 0.017$ was considered statistically significant after Bonferroni correction ($\alpha = 0.05/3 = 0.017$). Source data is provided as a Source Data file or available at https://doi.org/10.6084/m9. figshare.13136078.v1.

neurological symptoms such as lower limb paralysis after CAR T therapy, which was later identified as grade 4 ICANS after excluding other possible causes such as infection, Guillian–Barre syndrome, and leukemia involvement. In most clinical trials, the reported incidences for severe CRS (grade ≥ 4) were above 15%; with 13–63% of the patients experiencing neurotoxicity[19–22], including rare cases of fatal cerebral edema. The pathogenesis of CRS and neurotoxicity is incompletely understood, generally

considered in relation to the macrophages and endothelial activation induced by an immune response and released cytokine[20–23]. The lower incidences of severe CRS and neurotoxicity recorded in our trial may be due to the different CAR structure of the Sino 19 cell used, which expressed IgG4 Fc spacer with lower FcR-binding affinity, capable of reducing off-target FcR interaction while maintaining antitumor efficacy.

About efficacy, our trial achieved an 80.9% overall remission rate (CR/CRi) which is similar to the 81–85% recently published in phases I/II clinical trials, but the duration of response differ[11,12,16]. Maude et al. reported a 1-year RFS and OS rates of 59 and 76%, respectively in phase II, multicenter study using CD19–CD137 (4–1BB)-CD3z CAR T for children and young adults with r/r B-ALL[12]. Park et al.[11] showed that the median EFS and OS were 6.1 and 12.9 months, respectively in a phase I trial involving adults with r/r B-ALL using CD19-CD28-CD3z CAR T cell. Hay et al.[16] evaluated the outcome of phase I/II clinical trial for adults with r/r B-ALL by defined CD4$^+$:CD8$^+$ composition CD19-CD28-4–1BB -CD3z CAR T cells, the median EFS and OS were 7.6 and 20.0 months, respectively[6]. In our trial, the rates of RFS and OS at 1 year were 45 and 53%, and the median RFS and OS were 10.6 and 13.8 months, respectively. The difference in CAR constructs and manufacturing methods could influence the efficacy, however, other factors may also have played a role; our trial included more patients with EMDs and those with high-risk cytogenetic/molecular factors, so we mainly focused on investigating the patients' characteristics.

With regards to the initial response rate, we found that the CR/CRi rate in patients with EMDs other than CNS was the lowest, significantly lower than that in patients without EMDs or with CNS involvement only. This may be explained by the CAR T cell ability to cross the blood–CSF barrier to reach circulating tumor cells in the CSF to kill them, but hurdles restricting efficacy for the other EMDs may exist[23]. With respect to relapse after remission, our trial showed a 55% relapse rate within 1 year, a little higher than the 33–50% reported by recent studies[11,12,16]. Further analysis found that most patients relapsed after Sino 19 cell loss, but some patients who had EMDs, higher Tregs, and high-risk genetic/molecular factors relapsed during CAR T-cell persistence, only one had CD19-negative relapse. These imply that there are other factors related to the relapse in addition to the loss of CAR T cells or tumor antigens. As expected, Kaplan–Meier analysis showed that any subtype of EMDs, higher Tregs, and high-risk genetic factors were all significantly associated with poor OS and RFS, and although Tregs levels were positively correlated with EMDs, multivariable stepwise modeling analysis still demonstrated that EMDs and higher Tregs were independent high-risk factors respectively for death and relapse after CAR T-cell therapy. Our finding is in agreement with reports in other studies showing that leukemia patients with EMDs may have a more complex and stronger inhibitory immune microenvironment which consists of Tregs, myeloid-derived suppressor cells, and other varied components that play important roles in the initiation and progression of malignant tumor, including resistance to immunotherapy[24,25]. A xeno-transplant murine model study had shown that the means by which Tregs were modified to express CD19-targeted CARs in vitro could also efficiently inhibit the proliferation of activated human T cells, as well as the capacity of CAR T cells to lyse CD19-positive Raji tumor cells[26]. Here, we only analyzed the level of Tregs before infusion, whether the in vitro generation of Sino 19 cells can modify the surface structure of Tregs, and how the circulating Tregs in the blood interact with the Sino 19 cell need further exploration. For all that, we think that EMDs and related immunosuppressive microenvironment, such as higher Tregs are

harmful factors for CAR T efficacy, developing a modified CAR T model incorporating Tregs controls should be beneficial. Recently, some scholars proposed strategies and made preliminary researches on how to deplete Tregs or control their functions to improve CAR T cells engraftment in treating solid tumors[27–29]. In addition, Park et al.[11] reported that higher disease burden (≥5% BM blasts or EMD) was associated with shorter OS by univariable analyses, and Hay et al.[16] revealed some biomarkers (higher lactate dehydrogenase (LDH) concentration/lower platelet count pre-lymphodepletion), and treatment factors (incorporation of fludarabine into the lymphodepletion regimen/allo-HSCT after CAR T-cells infusion) were associated with a higher risk of relapse using a stepwise multivariable modeling. Generally, high LDH concentration and low platelet count are associated with tumor burden and an immunosuppressive tumor microenvironment/poor T-cell function[30–34], so from the perspective of patient factors, we think these results actually reflected the same essence. On treatment-related factors, due to the varying lymphodepletion regimen among most trials, we did not analyze the influence of the composition such as fludarabine on CAR T. Considering allo-HSCT is a standard therapy for patients with r/r ALL[27], we evaluated the effect of allo-HSCT on OS and RFS in a time-dependent Cox model; the results showed that patients who underwent allo-HSCT after CAR T had a lower risk of death and relapse, but not statistically significant when compared with those who did not. Hay et al.[16] demonstrated that patients undergoing allo-HSCT after CAR T therapy were significantly associated with better EFS by univariable and multivariable stepwise modeling analysis, conversely, Park et al.[11] found no significant difference in EFS and OS by log-rank test. Analyzing these results comprehensively, we think whether patients can derive benefit from HSCT after CAR T therapy need further definitive randomized studies to validate, but for patients who are eligible for transplantation, HSCT should be considered.

Philadelphia chromosome (Ph)$^+$, Ph-like ALL, t (v; 11q23), hypodiploidy, and complex karyotype (≥5 abnormalities) are the main cytogenetic and molecular abnormalities that contribute to the poor outcome of adult and pediatric patients with ALL[27]. In our trial, Kaplan–Meier analysis demonstrated that high-risk cytogenetic factor was associated with shorter OS and RFS, but it was not an independent predictor by multivariable analysis. This is consistent with other reports[11,16].

In brief, Sino19 cells produced a high remission rate in pediatric and adult patients with r/r B ALL. Some achieved longer RFS without additional therapy. EMDs and higher Tregs may be the important independent prognostic factors for poor outcome of CAR T therapy. For these special subgroups of patients, combining CAR T with other methods/therapies such as bridged allo-HSCT may be beneficial; additionally, more specific individualized therapies should be developed[35,36].

## Methods

**Clinical protocol design**. This is a single-arm, non-randomized, phase II clinical trial of Sino 19 cells in patients with CD19-positive r/r B-ALL designed at the Second Hospital of Anhui Medical University (SHAMU). The primary objective of the study was to assess the safety and overall remission rate of Sino 19 cell therapy. The secondary objective was to assess the OS, RFS, then explore their possible influence factors. Prespecified outcomes were as follows: adverse reactions above 4th grade are less than 10%; overall remission rate is about 70% (at least not less than 30%); OS, RFS, and possible influence factors are as results of exploration. The study protocol was first approved by the institutional review boards at SHAMU (the Medical Ethics Committee and the Academic Committee at SHAMU) on October 22, 2015. The accepted original protocol was deposited on the SHAMU institutional repository [http://www.ay2fy.com/kyb/chn_1157/content.jsp?id=8728]. An English translation of the main sections (including inclusion/exclusion criteria, and prespecified outcomes) of the original study protocol is available as Supplementary Note 1 within the Supplementary Information file. The first patient was enrolled on November 2, 2015 (Actual Study Start Date). We

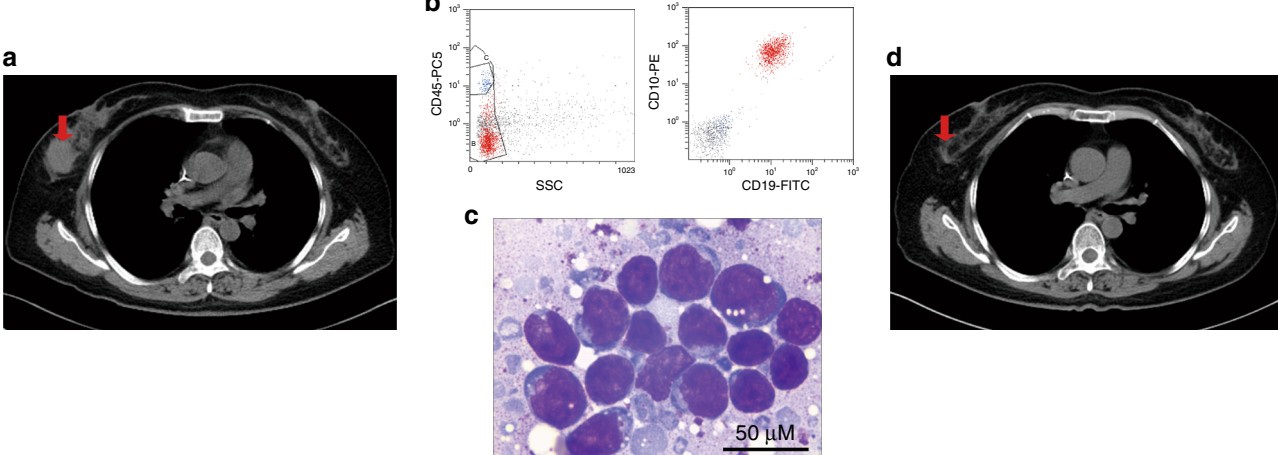

**Fig. 6 Diagnosis and therapeutic evaluation of EMDs before and after Sino 19 cell infusion.** B-ALL presented as a large breast mass in a patient with EMDs. **a** Computed tomography shows a large mass in the right breast (red arrow). **b** Flow cytometry on a breast mass sample acquired by puncture biopsy: shows CD19 and CD10 positive B lymphoblasts. **c** Cell morphology (Wright–Giemsa staining) shows lymphoblasts. At least three smears from this aspirated mass tissue were produced and stained, then three pathologists independently performed morphological view and evaluation, the final diagnosis was established when they had the consistent findings. Scale bar: 50 μM. **d** Computed tomography shows disappeared large mass at Day 50 following Sino 19 cell infusion (red arrow).

started the proceedings for registration on January 25, 2016 at www.clinicaltrials.gov and this was completed on April 12, 2016 as NCT02735291. In the meantime, 10 patients were enrolled in the trial but once registration was complete, an additional 41 patients were enrolled with the last patient recruited on May 13, 2019. We advertised the trial on the SHAMU website, at local medical meetings, and scientific conferences. Before enrolling in this study, patients or their guardians were provided written informed consent according to the Declaration of Helsinki.

**Patients screening**. After the patients signed the informed consent, we collect the demographics, previous clinical data and did some related examinations if necessary; these included ages, gender, weight, height, date of diagnosis, cycles of chemotherapy, cytogenetic or molecular abnormalities, imaging, and previous allo-HSCT, etc. CNS involvement was defined as blast cells detected in CSF (CSF sample with ≥5 leukocytes per cubic millimeter and <10 erythrocytes per cubic millimeter). EMDs were defined as CD19+ B lymphoblastic cells outside the BM, detected by revealing abnormal masses in extramedullary sites using imaging examinations, such as magnetic resonance imaging, computed tomography, or positron-emission tomography and cytological or pathological biopsy examination to detect blast cells in these masses (Fig. 6). Then we further identify the patients who could receive CAR T treatment according to the eligibility criteria in the study protocol (Supplementary Note 1).

**Preconditioning chemotherapy**. Three chemotherapy regimens were used including fludarabine (FC), VDCP, and Cy. FC (fludarabine 25 mg m$^{-2}$ followed by cyclophosphamide 400 mg m$^{-2}$ × 5 days) was used if a patient had normal white blood cell count and active BM hyperplasia; VDCP (Vincristine 1 mg m$^{-2}$ + Daunorubicin 60 mg m$^{-2}$ + Cyclophosphamide 600 mg m$^{-2}$) × 1 day + Prednisone 2 mg kg$^{-1}$ × 7 days then followed by FC × 3 days, if a patient had higher tumor burden such as percentage of marrow blasts ≥50% or EMD positive; Cy (Cyclophosphamide, 400 mg m$^{-2}$ × 3 days), if a patient had leukopenia and hypoplastic marrow.

**Sino 19 cell infusion and follow up**. A single dose of Sino 19 cells infused to the eligible patients ranged 1–5 × 10$^6$ transduced viable T cells per kilogram (kg) of body weight for each course of treatment; the total dose of viable T cells did not exceed 2 × 10$^9$. In the beginning, 1/10 of the total cells were administrated on Day 0, if there was no obvious adverse reaction, the rest were infused on Day 1. Administration of lymphodepletion/bridging chemotherapy was completed 3–14 days before the start of Sino 19 cell infusion to induce lymphopenia or reduce tumor load to facilitate engraftment and homeostatic expansion of Sino19 cells. Bone marrow puncture and its related examinations were conducted on day 14–28 post infusion to evaluate the response and remission status. Between day 30 to 180, observation and monitoring on the BM/PB and EMDs were conducted monthly and subsequently conducted every 3 months. During follow up, a BM aspiration examination was immediately taken if the patient has any suspicious symptoms of relapse. The other tests such as the persistence of CAR T cells, lymphocyte subsets, etc. were all regularly performed according to the preset in the study protocol.

**Assessment of efficacy**. CR/CRi was defined according to the National Comprehensive Cancer Network guidelines, version 1, 2015[37]. OS was calculated from the date of Sino 19 cell infusion to the date of death. RFS was from confirmation of remission to relapse. MRD negative was defined as an absence of immunophenotypically abnormal blasts in the PB/BM by multiparametric flow cytometry (limit of detection 1:10,000).

**Assessment and management of toxicities**. The degree of toxicities was in accordance with the National Cancer Institute Common Terminology Criteria for Adverse Events (CTCAE v4.03)[38]. We diagnosed and graded the CRS and ICANS according to the American Society for Blood and Marrow Transplantation consensus grading system when we retrospectively analyzed the data[39].

**CAR design and construction**. The construct of the Sino19 CAR vector in this study was designed as follows: retrovirus long terminal repea, CD19-specific scFv (CD19-scFv), hIgG4-CH2CH3, CD28TM (transmembrane domain), CD137, and an intracellular signaling domain of CD3ξ. The single-chain (scFv) against human CD19 was derived from anti-CD19 monoclonal antibody (mAb) FMC63[40]. The scFv codon sequence was optimized and synthesized by GenScript (Nanjing) Co., Ltd. and cloned in frame with the human IgG4-CH2CH3 domain, CD28 transmembrane domain, and cytoplasmic signaling domain, CD137 cytoplasmic signaling domain, and with the ζ chain of the TCR/CD3 complex in the SFG retroviral backbone. The human IgG4-CH2CH3 domain contains the hinge and CH2CH3 domain derived from 104 to 327 of UniProtKB/Swiss-Prot P01861.1. CD28 transmembrane domain and cytoplasmic signaling domain were derived from 69 to 136 of NCBI reference sequence XP011510499.1. CD137 (4-1BB) cytoplasmic signaling domain was derived from 214 to 254 of NCBI reference sequence NP001552.2. The ζ chain of the TCR/CD3 complex was derived from 51 to 163 of NCBI reference sequence NP000725.1. Thus, both CD28 and CD137 (4–1BB) co-stimulatory domains were included in the CAR constructs (Supplementary Fig. 1a).

**Manufacturing of Sino 19 cells**. All the Sino 19 cells were derived from autologous T cells and manufactured in Sinobioway Cell Therapy Co., Ltd., China. Totally, 80–120 mL of peripheral blood (PB) was drawn from the patient and immediately sent for PB mononuclear cells (PBMC) separation or cryopreserved in liquid nitrogen if it cannot be processed immediately to prepare the Sino 19 cells. Isolated PBMC were cultured with anti-CD3/CD28 antibody (Miltenyi Biotec, Germany), in a culture medium containing interleukin-2 (Sihuan Pharm, Beijing) for activation and enrichment of T cells. The retroviral vector was added at day 3 of cell activation and was washed out after 1 day of culture. Cells were then incubated in the cell incubator (Thermo, USA) for further expansion for 10–12 days. The amounts of total cells and CAR T-positive rates were monitored constantly during the entire culture process period. On the final day of culture, the modified T cells were washed and harvested into an infusion bag with 100 mL saline containing moderate human serum albumin. The acceptable assessment criteria for the final products (infusible cells) included: cell viability ≥ 90%, CD3+ cell ≥ 90%, endotoxin ≤ 0.5 EU L$^{-1}$, mycoplasma negative, bacterial, and fungal cultures negative, and target gene transfection rate or expression ≥20% (Supplementary Fig. 1b).

To assess the in vitro cytotoxic function of the generated final Sino 19 cell products, we evaluated the final products derived from two patients with r/r B-ALL at the initial stage of the trial. The Sino 19 cells were cocultured with the patients' respective CD19+ B lymphoblasts isolated from their BMs (1:1 ratio) for 24 h. The results show that the manufactured Sino 19 cells could target kill the lymphoblasts in a CD19 specific manner (Supplementary Fig. 1c, d).

**Proliferation and persistence assessment of Sino 19 cells in vivo.** For Sino 19 cell expansion and persistence assessment in vivo, genomic DNA was isolated directly from samples of whole blood obtained before and after infusion of Sino 19 cell, some patients' BM and CSF samples were used if possible. Qualitative and quantitative real-time polymerase chain reaction (Q-PCR and qRT-PCR) assays were performed to detect the integrated CD19 CAR DNA copies using LightCycler 96 System (Roche, USA) with a primer for the CD137-CD3ζ junctional region which was synthesized by Sangon Biotech (Shanghai) Co., Ltd. An eight-point standard curve was generated using the CD19 CAR plasmid and the CAR DNA copies of the patients were calculated from the standard curve (LightCycler 96 Application Software 1.1). In order to determine whether the Sino 19 cell has expanded, the copies of CAR DNA were detected 2 h after infusion as a baseline value by qRT-PCR, then whenever the patients had the first fever, and every 3–5 days for 3 weeks, then about every 2 months until it can not be detected. The horizontal line at five copies per microgram of DNA represented the lower limit of quantification of this assay.

**Detection of Tregs.** T cell subtypes including Tregs of all patients before treatment were evaluated with the method reported by our research team[41,42]. Two to five millilitre of PB was collected from each patient. All samples were anti-coagulated with heparinized tubes and examined within 4 h. Hundred microlitre PB was incubated at 25 °C for 15 min with the specific mAbs, which included FITC-CD25-mAb (B1.49.9 clone), PE-CD127-mAb (R34.34 clone), PC5.5-CD4 mAb (13B8.2 clone), and the appropriate isotype controls. After incubation, red blood cells were lysed and quickly assessed by flow cytometer FC-500 and analyzed using the CXP 2.0 Software (Beckman Coulter, USA). All mAbs specific for human surface antigens were purchased from Beckman Coulter-Immuno tech (Marseille, France). The level of Tregs was expressed as a percentage of CD4+CD25+CD127low T cells in CD4+ T cells by sequential gating on lymphocytes and CD4+ T cells (Supplementary Fig. 2). In our laboratory, the 99% confidence interval of mean Tregs in a healthy population is 5.28–5.94, we set the upper limit as a cutoff value, and then classified the patients who were evaluated for efficacy into higher Tregs group (>5.94) and normal or lower Tregs group (≤5.94).

**Statistics and reproducibility.** Categorical variables were presented as sample size and percentage (n (%)) and compared using the chi-squared test (or Fisher's exact test). Kaplan–Meier approach was performed to estimate time-to-event analysis, and the log-rank test was used to evaluate between-group differences in survival curves. Multivariable analysis (EMDs, Tregs, and high-risk cytogenetics) was compared using the Cox proportional hazard model. In addition, the time-dependent Cox model was used to compare the differences between patients with and without a bridged allo-HSCT group. Continuous variable with normal distribution was compared by one-way ANOVA, and multiple comparisons between groups were compared using LSD tests. All statistical analyses were performed by Statistical Product and Service Solutions (SPSS) 23.0. A two-tailed $p$ value < 0.05 was considered significant statistical analysis, and Bonferroni adjustment of $p$ value for multiple testing was calculated as $0.05/n$ (where "$n$" indicates the number of comparisons). The statistical power was calculated by using Power Analysis and Sample Size (NCSS-PASS) 11.0, the significant level (alpha) was 0.05.

In this study, the identification of blast cells in PB smear, or the aspirate smears from BM and mass tissue, or the biopsy slides were all evaluated by three pathologists independently, the final diagnosis report was based on their consistent findings. Counting the blast cells percentage to all nucleated cells in each blood or BM smear was repeated at least twice.

## Data availability

The source data underlying Figs. 1–3, 4a, c, 5, and Tables 1–3, as well as the other individual de-identified participant's data that support the findings of this study, are provided with this paper as a Source Data file or available at https://doi.org/10.6084/m9.figshare.13136078.v1. The original study protocol is accessible at http://www.ay2fy.com/kyb/chn_1157/content.jsp?id=8728, or from the administrator (wangjiyu1992@126.com) and the corresponding authors upon request. An English translation of the main sections of the Study Protocol is available in the Supplementary Information file. The remaining data is available within the Article, Supplementary Information, or available from the authors upon request. Source data are provided with this paper.

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

## Acknowledgements
The sponsor of this trial was Sinobioway Cell Therapy Co., Ltd., which was in charge of trial registration, CAR T cells manufacture, and detection by PCR. They were involved in the study design but had no role in patient recruitment, clinical data collection, and analysis, the decision to publish, or preparation of the paper. The staff of the Hematologic Diseases Research Center of Anhui Medical University assisted with clinical care and some examinations. We thank PhD Xiaotong Song for his contributions to the trial design and cell quality control, thank Alice Charwudzi for helping with the paper review. We thank PhD Xu Zhang for double-checking our statistical analyses. We also thank the following hospitals for recommending r/r B-ALL patients to join this trial: The First Hospital of Anhui Medical University, the First Affiliated Hospital of USTC (University of Science and Technology of China), Suzhou Municipal Hospital, and the Second Hospital of Bengbu Medical College. Funding was provided by the National Natural Science Foundation of China (No. 81670179); Spark Research Fund of SHAMU (No. 2015hhjh03); University Academic Technology Leaders Subsidized Projects of Anhui Province (No. 2014D028); Major Science and Technology Project of Anhui Province, China (No. 201903a07020030).

## Author contributions
The first author wrote the first draft of the paper in conjunction with the other coauthors. All the authors discussed and interpreted the study results and vouch for the accuracy and completeness of the data and analyses. Zhimin Zhai, Zhenqing Feng, and Zhenyun Liu designed the trial and the structure of CAR. Zhenqing Feng and Furun An were responsible for statistical tests and analysis. Zhenyun Liu, Huiping Wang, and Yanjie Ruan were responsible for the production and the quality monitoring of Sino 19 cells. Huiping Wang and Yanjie Ruan were in charge of MRD and Tregs detection by flow cytometry. Furun An, Fan Wu, and Jiakui Zhang collected the source data. Zhimin Zhai and Furun An wrote the final paper. Yongling Fu was in charge of trial registration at ClinicalTrials.gov and harmonize between clinical and laboratory. The other authors assisted with clinical care, data verification, and results in confirmation.

## Competing interests
The authors declare that we have no financial and personal relationships with other people or organizations that can inappropriately influence our work. There is no professional or other personal interest of any nature or kind in any product, service, and/or company that could be construed as influencing the position presented in, or the review of, the paper entitled. We declare that no competing interests exist regarding the publication of this paper.
