## [Peer Review File · Nature Communications]

Reviewers' comments:

Reviewer #1 (Remarks to the Author):

The manuscript describes the outcomes of 47 patients with R/R AML treated with CD19 targeted CAR T cells. They report response rates and overall survival comparable to studies previously published. The authors then retrospectively evaluate patient characteristics which predict for responses and durability of responses. Using these retrospective analyses, since the phase I clinical trial was not designed to address these variables, the authors found that extramedullary disease, poor genetic prognostic factors, and Tregs all correlated to poorer overall survival. Overall, this report is largely confirmatory of multiple other previously published CAR T cell/ALL studies and is largely descriptive. There are multiple problems with this manuscript.

1. The authors fail to describe the CAR construct, this is relevant and should have been added.
2. There is a complete lack of mechanistic insight into why these variables could or in fact do lead to poorer outcomes.
3. The Treg data is minimal. Were there more Tregs in the CAR T cell product? Did patients with increased Tregs have this as well after conditioning chemotherapy?
4. The fact that EMD patients had increased Tregs, is this a common finding? The fact that these 2 variables correlate, makes one wonder if these are indeed 2 independent variables.
5. The definition of CR lacks any data on MRD analyses, why is this the case?
6. The conditioning regimens differ between patients, this is certainly a very strong confounding variable.
7. The authors report on B cell aplasias and CAR T cell persistence. This is known, what would have been of interest is whether either of those variables relate to patient outcomes.
8. The authors do not report on CD19- tumor relapses, this should have been included.
9. With respect to EMD, why did the authors not biopsy persistent tumors? This would have been extremely informative especially given the question of Tregs playing a role in poorer outcomes.
10. The manuscript is very poorly written, the authors needs to have an editor review the manuscript.

Reviewer #2 (Remarks to the Author):

Dear Dr. An and colleagues,

I had the pleasure of reviewing your manuscript, "Influence of Patient Characteristics on CAR T-Cell Therapy in ALL."

Comments are as follow:

1. General: Please obtain a review of the manuscript as a whole by someone who has mastery of written English. There are multiple grammatical and typographical errors through out the manuscript that are minor, but warrant editing. (Examples: spelling mistake: CD19-posotive, should be positive; example, "the median time were" should be " the median time was." Another example, 'except 2 aged patients,'--I do not understand what this means. Tosilizumab is spelled incorrectly
2. General: CNSL is not common nomenclature. Please use CNS involvement in the abstract.
3. Methods: How was CRi defined. What parameters were used to determine incomplete count recovery.
4. Statistical analysis: Please clarify when t-test was used vs. Mann Whitney, the latter of which

would be preferred given the small number of patients and likely non-parametric data.

5. Results: Please clarify when and why steroids were used preferentially over tocilizumab

6. Results: Please clarify additional details of the patient with progressive paralysis. Was there spinal cord compression or leukemia involvement?

7. Methods: Please add details about the co-stimulatory domain to the manuscript.

8. Results: What information do you have regarding Tregs and how this was impacted by prior therapies.

9. Results: it is mentioned that TKIs could be used for Ph+ ALL, but recent data supports that TKI may modulate CAR persistence. What decision making was employed when adding TKI

10. Re: EMD disease, how was this routinely monitored.

11. Results: did anyone look at the impact of prior blind?

Responses and explanations to Reviewer #1 comments:

1. The authors fail to describe the CAR construct, this is relevant and should have been added.

We agree with the reviewer. The description of the CAR construct has been added to the Supplemental Material text, titled as 'CAR design and construction' section on page ---. This section reads as follows:

CAR design and construction

The construct of Sino19 CAR vector in this study was designed as follows: LTR (retrovirus long terminal repeat), CD19-specific scFv (CD19-scFv), hIgG4-CH2CH3, CD28TM (transmembrane domain), CD137, and CD3 ζ . The single chain (scFv) against human CD19 was derived from anti-CD19 monoclonal antibody FMC63 (Zola, Immunol cell biol. 1991, Dec 69). The scFv codon sequence was optimized and synthesized by GenScript(Nanjing) Co. Ltd and cloned in frame with the human IgG4-CH2CH3 domain, CD28 transmembrane domain and cytoplasmic signaling domain, CD137 cytoplasmic signaling domain, and with the ζ chain of the TCR/CD3 complex in the SFG retroviral backbone. The human IgG4-CH2CH3 domain contains the hinge and CH2CH3 domain derived from 104-327 of UniProtKB/Swiss-Prot P01861.1. CD28 transmembrane domain and cytoplasmic signaling domain was derived from the 69-136 of XP011510499.1. CD137 (4-1BB) cytoplasmic signaling domain was derived from the 214-254 of NCBI reference sequence NP001552.2. The ζ chain of the TCR/CD3 complex was derived from the 51-163 of NCBI reference sequence NP000725.1. Thus both CD28 and CD137 (4-1BB) costimulatory domains were included in the CAR constructs.

2. There is a complete lack of mechanistic insight into why these variables could or in fact do lead to poorer outcomes.

This study is a single-arm, phase 2 clinical trial, so the emphasis was mainly on safety and complete remission rate, we then further analyzed the correlation between patient characteristics and efficacy according to the study protocol. The mechanistic insight should be very complex; we will explore this further in our future research.

3. The Treg data is minimal. Were there more Tregs in the CAR T cell product? Did patients with increased Tregs have this as well after conditioning chemotherapy?

We agree with the reviewer's concern, indeed the Treg data was inadequate. The Tregs data shown in our manuscript was just the levels analyzed after conditioning chemotherapy and before CAR T cell infusion. In fact, we also tested the Tregs in the CAR T cell products and at different time points after CAR T cell infusion. We did not find more Tregs in the CAR T cell products. There are lots of published data about Tregs, however, our research team is currently undertaking in-depth investigation on Tregs, and we hope to make a systematic analysis on this for a special report in our next publication. This has been captured in our revised manuscript.

4. The fact that EMD patients had increased Tregs, is this a common finding? The fact that these 2 variables correlate, makes one wonder if these are indeed 2 independent variables.

Thank you for these remarks. In our study, all 13 (46.6%) patients with extramedullary diseases (EMDs) as against 15 (53.6%) patients without EMDs at baseline recorded high Tregs levels ($P = 0.001$).

	EMD+	EMD-	Total	P value
Tregs >5.94	13(46.6%)	15(53.6%)	28	0.001
Tregs \leq 5.94	0	18(100.0%)	18	

Based on several clinical data reports, increased circulating Tregs are known to be associated with a high tumor infiltration of tissues/extramedullary sites in various types of cancers including ALL; and have been shown to be a contributor to poorer prognosis¹⁻⁵. However, there are limited reports about Tregs in EMD patients. Nonetheless, the

higher Tregs in all patients with EMDs suggests that EMDs may behave like other solid tumor tissues, and together with the higher Tregs may induce a stronger immunosuppressive microenvironment that can probably interfere with CAR T cell efficacy^{6,7}. Most of these explanations have been added to the revised manuscript.

Related references on this issue are as follows:

- 1) Ohue Y, Nishikawa H. RegulatoryT (Treg) cells in cancer: Can Treg cells be a new therapeutic target? *CancerSci*. 2019 Jul; 110(7): 2080-2089.
- 2) M. Niedźwiecki, O. Budziło, E. Adamkiewicz-Drożyńska, et al. CD4+CD25highCD127lowFoxP3+ Regulatory T-Cell Population in Acute Leukemias, A Review of the Literature, *J Immunol Res*. 2019; Published online 2019 Mar 3.
- 3) M. Niedźwiecki, O. Budziło, M. Zieliński, et al. CD4+CD25^{high}CD127^{low/-} FoxP₃⁺ Regulatory T Cell Subpopulations in the Bone Marrow and Peripheral Blood of Children with ALL: Brief Report, *J Immunol Res*. 2018; Published online 2018 May 29.
- 4) J Duell, M Dittrich, T Bedke, et al. Frequency of regulatory T cells determines the outcome of the T-cell-engaging antibody blinatumomab in patients with B-precursor ALL, *Leukemia*. 2017 Oct; 31(10): 2181–2190.
- 5) Takeuchi Y, Nishikawa H. Roles of regulatory T cells in cancer immunity. *Int Immunol*. 2016 Aug; 28(8): 401-409.
- 6) Lim WA, June CH. The principles of engineering immune cells to treat cancer. *Cell*, 2017; 168(4): 724–740.
- 7) Shum T, Kruse RL, Rooney CM. Strategies for enhancing adoptive T-cell immunotherapy against solid tumors using engineered cytokine signaling and other modalities. *Expert Opin Biol Ther*, 2018; 18(6): 653-664.

We agree with the reviewer’s concern about the correlation between EMDs and higher Tregs levels, we also wondered if these are two independent variables but due to the small numbers of EMD cases, we could not perform multivariate analysis to control for confounding factors. However, based on this good remark from the reviewer, we analyzed the data using Cox regression analysis. The results showed that active EMD before infusion was an independent factor for OS (P=0.004) and a higher risk of death, HR=0.220 (95% CI 0.077-0.623); higher Tregs level before infusion was also an independent factor for RFS (P=0.010) and is associated with a higher risk of relapse, HR=0.196 (95% CI 0.057-0.676). The Details of the analysis is presented in the table below; we have also added the results to the revised manuscript in the methods, results and discussion sections.

Tab.1 Multi factor analysis for OS

		Variables in the Equation						95.0% CI for Exp(B)	
		B	SE	Wald	df	Sig.	Exp(B)	Lower	Upper
Step 1	EMD+	-1.515	.532	8.120	1	.004	.220	.077	.623

Tab.2 Multi factor analysis for RFS

		Variables in the Equation						95.0% CI for Exp(B)	
		B	SE	Wald	df	Sig.	Exp(B)	Lower	Upper
Step 1	higher Tregs	-1.628	.631	6.656	1	.010	.196	.057	.676

5. The definition of CR lacks any data on MRD analyses, why is this the case?

The National Comprehensive Cancer Network (NCCN) Clinical Practice Guidelines in Oncology: Acute Lymphoblastic Leukemia, Version 1.2015 was the criteria used for our responses. However, in our study, the detection of MRD was regularly performed for all patient before and after CAR T infusions, this was described in the methods and results

sections in our previous manuscript, the sentences read: (Minimal residual disease (MRD) negative was defined as an absence of immunophenotypically abnormal blasts in the peripheral blood (PB)/bone marrow (BM) by multiparametric flow cytometry (limit of detection 1:10,000). Among 38 CR/CRi patients, 1 was positive for MRD (0.29%) and this patient relapsed 180 days following infusion).

6. The conditioning regimens differ between patients, this is certainly a very strong confounding variable. The conditioning regimens aimed at lymphodepletion and possibly to enhance CAR T cell expansion and persistence in vivo. Up till now, no single conditioning regimen has been approved as an optimal regimen for enhancing CAR T cell efficacy in all patients; various clinical trials used conditioning regimens that varied between patients. Our trial enrolled high-risk r/r ALL patients who were heavily pretreated with intensive chemotherapy with/without alloHSCT before CAR T treatment; hence the low dose conditioning regimens used based on the patient's condition should not have a significant effect on the overall efficacy of these patients. In fact, the conditioning regimens in different treatment centers and even within the same centers differ. We adopted the conditioning regimens reported by Maude SL team at the University of Pennsylvania. We have included excerpts from some published articles for your kind perusal below:

- 1) Excerpt from Connie et al. (Connie Lee Batlevi, Eri Matsuki, Renier J. Brentjens, Anas Younes, Novel immunotherapies in lymphoid malignancies, Nat Rev Clin Oncol. 2016 Jan; 13(1): 25–40.)

[Redacted]

- 2) Excerpt from Supplementary Appendix of Maude et al. (Maude SL, Frey N, Shaw PA, et al. Chimeric antigen receptor T cells for sustained remissions in leukemia. *N Engl J Med*, 2014; 371: 1507-1517.)

[Redacted]

7. The authors report on B cell aplasia and CAR T cell persistence. This is known, what would have been of interest is whether either of those variables relates to patient outcomes.

We thank the reviewer for this positive assessment. In order to respond to this comment, we carefully reviewed related literatures. Generally, longer durability of the CAR T-cell engraftment is considered to improve upon CAR T-cell therapy and others also associated the disease relapse to a lack of CART cell persistence¹⁻³, but the optimal length of CART cell persistence remains unknown^{4,5}. Our data seems to agree with these findings. This comment is addressed in the revised 'Persistence of Sino 19 cell' section in the following sentences:

In total, 19 patients relapsed, 78.9% (15/19) happened after Sino 19 cell loss; the other 4 (21.1%) relapsed whilst Sino 19 cell was still persisting in vivo which included one CD19 negative relapse, these 4 patients all had EMDs, abnormally increased Tregs, and high risk genetic abnormalities simultaneously. For patients who maintained their remission up to the data cut-off time, 66.7% (6/9) lost Sino 19 cells. Interestingly, Sino 19 cells and B cell aplasia were detected in 9 unresponsive/refractory patients within 60 days post infusion; most of them (89%, 8/9) had either EMDs (1 with CNS involvement only, 5 with multi lesions other than CNS) or high risk cytogenetic/molecular abnormalities (2 positives for E2A/PBX1), the 9th patient had no EMDs but her cytogenetic data was not acquired. Sino 19 cells were successfully detected in only 7 patients CSF samples which included 3 patients with CNS involvement from day 28 to day 180 after infusion (generally, nucleated cells in CSF is lower), all of them achieved remission. Due to the small number of patients and limited observation time, we could not further analyze the relationship between Sino 19 cell persistence or B cell aplasia and patient outcomes. Analysis of further cases is necessary in future.

Related references on this issue are as follows:

- 1) Davila, M. L. et al. Efficacy and toxicity management of 19-28z CAR T cell therapy in B cell acute lymphoblastic leukemia. *Sci. Transl. Med.* 2014; 6: 224ra25.
- 2) Maude, S. L. et al. Chimeric antigen receptor T cells for sustained remissions in leukemia. *N. Engl. J. Med.* 2014; 371: 1507–1517.
- 3) Lee, D. W. et al. T cells expressing CD19 chimeric antigen receptors for acute lymphoblastic leukaemia in children and young adults: a phase 1 dose-escalation trial. *Lancet*, 2015; 385: 517–528.
- 4) Battlevi CL, Matsuki E, Brentjens RJ and Younes A. Novel immunotherapies in lymphoid malignancies. *Nat Rev ClinOncol.* 2016; 13(1): 25-40.
- 5) Tomuleasa C, Fuji S, Berce C, et al. Chimeric Antigen Receptor T-Cells for the Treatment of B-Cell Acute Lymphoblastic Leukemia. *Front. Immunol.* 2018; 9: 239-253.

8. The authors do not report on CD19- tumor relapses, this should have been included.

We thank the reviewer for bringing this to our attention. At data cutoff, there was one patient with CD19 negative relapse (CD19 negative on surface of blast cells). This patient was [Redacted] who had Ph-like (IgH/EPOR) rearrangement gene. In response to this suggestion, we revised the results section and addressed it under 'Persistence of

Sino 19 cell'.

9. With respect to EMD, why did the authors not biopsy persistent tumors? This would have been extremely informative especially given the question of Tregs playing a role in poorer outcomes.

At first sight, we agreed with this suggestion. However, most of the patients with relapsed and refractory leukemia have received repeated and multiple intensive chemotherapies or even transplantation before enrollment. They are very weak/frail, often presenting with leucopenia and thrombocytopenia. So they cannot endure persistent pathological biopsy. Generally, we routinely confirm the extramedullary diseases through clinical manifestations in combination with fine-needle aspiration for cytology and molecular studies, and imaging examinations. We agree with the reviewer that, it would be very important to understand the role of Tregs via the biopsy of persistent extramedullary diseases (tumors). This would require further expansion of the study cohort to include more patients that can withstand repeated biopsies; we hope to include this in our future research or to further study the interfering mechanism of Tregs on the therapeutic effect of CAR T by animal model experiment.

10. The manuscript is very poorly written, the author's needs to have an editor review the manuscript.

We agree with the reviewer and appreciate the constructive comment and suggestion. We have carefully reviewed the whole manuscript. The manuscript has also been reviewed by English editors.

Responses and explanations to Reviewer #2 comments

1. General: Please obtain a review of the manuscript as a whole by someone who has mastery of written English. There are multiple grammatical and typographical errors throughout the manuscript that are minor, but warrant editing. (Examples: spelling mistake: CD19-positive, should be positive; example, "the median time were" should be "the median time was." Another example, 'except 2 aged patients' --I do not understand what this means. Tosituzumab is spelled incorrectly.

We agree with the Reviewer's constructive suggestions. We have carefully reviewed the whole manuscript and corrected the multiple grammatical and typographical errors. The manuscript has also been reviewed by English editors to correct these errors.

2. General: CNSL is not common nomenclature. Please use CNS involvement in the abstract.

Thank you for bringing this to our attention. We have revised the CNSL to CNS involvement in the abstract as well as in the main manuscript.

3. Methods: How was CRi defined? What parameters were used to determine incomplete count recovery?

Admittedly, the definition for CRi (complete remission with incomplete hematologic recovery) was not stated in the previous version manuscript. However, the National Comprehensive Cancer Network (NCCN) Clinical Practice Guidelines in Oncology: Acute Lymphoblastic Leukemia, Version 1.2015 was the criteria used for our responses. The revised version of the manuscript reads: 'CR/CRi was defined according to the National Comprehensive Cancer Network (NCCN) guidelines, version 1.2015⁽¹⁴⁾' under the heading 'Assessment of efficacy'.

The parameters used to determine CRi in accordance with the NCCN guidelines were as follows:

- 1) No circulating blasts or extramedullary disease.
- 2) Trilineage hematopoiesis (TLH) and <5% blasts.
- 3) Recovery of platelets but <100,000/microL or ANC <1000/microL.
- 4) No recurrence for 4 weeks.

4. Statistical analysis: Please clarify when t-test was used vs. Mann Whitney, the latter of which would be preferred given the small number of patients and likely non-parametric data.

This was not well clarified in our previous manuscript. In response, the Mann-Whitney U test was not used in this study. This has now been dealt with in the revised statistical analysis section as:

In this study, Chi-squared Test and Pearson correlation coefficient were used to evaluate the correlation between different factors (Table 2). The Kaplan-Meier approach was performed to estimate time-to-event analyses, log-rank tests were used to detect between-group differences in survival curves (Figure 2 and 3), and multivariate analyzed by Cox proportional hazard model. The correlation between two groups of variables was analyzed by linear regression analysis (Figure 5C).

For quantitative data analysis (Figure 6), the normality test and homogeneity of variance test were performed, the results showed that data of three groups accorded with normal distribution and homogeneity of variance, one-way ANOVA with Least Significant Difference (LSD) T test was then performed for comparison between groups.

5. Results: Please clarify when and why steroids were used preferentially over tocilizumab.

The pathophysiology of the immune effector cell-associated toxicities such as CRS and ICANS are unclear, most CD19 CAR T cell clinical trials found marked inflammatory cytokine especially IL-6 elevations in association with the toxicities, but the other cytokines such as tumor necrosis factor α , interferon γ , interleukin 10 (IL-10), IL-1 and so on may also play important roles⁽¹⁻³⁾. Initially, there were concerns about steroids inhibiting CAR T activity and expansion;

hence some scholars did not recommend its use in CART therapy⁽⁴⁾. But systemic corticosteroids are effective in dampening CRS due to its established broad-spectrum anti-inflammatory properties, especially for patients with high-grade CRS or ICANS⁽¹⁻³⁾. So when patients express severe symptoms such as continuous high fever with hypotension, hypoxia etc. (CRS \geq 3 degrees), we use some steroid (methylprednisone 0.5-1 mg / kg body weight per day about 3 days) as first line therapy after excluding other factors, if the symptoms do not improve and the level of IL-6 exceed the baseline value by more than 100 times, we then administer tocilizumab concomitantly.

Additionally, at the 2019ASH Annual Meeting, recent researches demonstrated that the early use of steroid reduce the rate of CART T cell related CRS and NEs without any meaningful clinical impact on the efficacy⁽⁵⁾.

Related references on this issue are as follows:

- 1) Neelapu SS, Tummala S, Kebriaei P, et al. Chimeric antigen receptor T-cell therapy assessment and management of toxicities. *Nat Rev ClinOncol*. 2018; 15(1): 47-62. Davila ML, Riviere I,
- 2) Lee DW, Santomaso BD, Locke FL, et al. ASBMT consensus grading for cytokine release syndrome and neurological toxicity associated with immune effector cells. *Biology of Blood and Marrow Transplantation*, 2019; 25(4): 625-638.
- 3) Murthy H, Iqbal M, Chavez JC, Kharfan-Dabaja MA. Cytokine Release Syndrome: Current Perspectives. *Immunotargets Ther*. 2019;8:43–52. doi:10.2147/ITT.S202015
- 4) Davila ML, Riviere I, Wang X, et al. Efficacy and toxicity management of 19-28z CAR T cell therapy in B cell acute lymphoblastic leukemia. *SciTransl Med*. 2014; 6(224): 224ra25
- 5) Topp M, Meerten TV, Houot R, et al. Earlier Steroid Use with Axicabtagene Ciloleuce (Axi-Cel) in Patients with Relapsed/Refractory Large B Cell Lymphoma , Oral report, <https://ash.confex.com/ash/2019/webprogram/Paper126081.html>

6. Results: Please clarify additional details of the patient with progressive paralysis. Was there spinal cord compression or leukemia involvement?

The patient with progressive paralysis was [redacted], diagnosed as Ph-positive acute B lymphoblastic leukemia on January 20, 2017 and treated according to the guidelines of Chinese adult Acute Lymphoblastic Leukemia cooperative Group. The course and efficacy evaluation for the patient are as shown as follows in the table:

Date	Treatment/Examination	Efficacy evaluation
2017-01-26	VDCP induction (vincristine sulfate, doxorubicin hydrochloride, cyclophosphamide, and dexamethasone)	
2017-02-20	BM and CSF examination	Achieved first CR, CR1 (MRD negative by FCM and RT-PCR)
2017-03-13to 2018-07-22	High dose methotrexate ×3 courses MA (Mitoxantrone and Ara-C) ×2 courses High dose Ara-C ×2 courses CAM (cyclophosphamide, Ara-C and 6-MP) ×2 courses VDCP ×1 courses	Continuous CR
In the interval without intensive chemotherapy, imatinib mesylate orally was used for maintenance treatment, intrathecal injection (IT) methotrexate ×8 times to prevent CNS involvement.		
2018-09-17	BM examination	Found 54% CD19+ blast cells (BC) in BM Relapse 1
2018-09-28	VDCP induction again	no CR, and complicated with severe infection during chemotherapy
2018-10-31	Blood draw and in vitro preparation of CAR T cells (Sino 19 cells)	
2018-11-09	FC ×3 courses, lymphodepleting chemotherapy	

2018-11-13	BM and CSF examination before CAR-T cell infusion (lumbar puncture without methotrexate IT)	Found 43% CD19+ BC in CSF; CNS involvement 16%CD19+ BC in BM
2018-11-14	CAR-T cells infused (Total number 1×10^9)	Fever, transient hypotension and hypoxia. CRS grade 3
2018-12-03 (Day 19)	BM examinations	BC in BM negative
2018-12-17 (Day 33)	BM, CSF and other indicated examinations	BC and MRD in BM and CSF all negative, CR 2
2018-12-20 (Day 34)	Head MRI, due to weakness and numbness of lower limbs	No abnormalities found
2018-12-23 (Day 38)	Spinal MRI, because weakness and numbness progressed, expressed paralysis	No occupying lesion/mass or other abnormalities found.
2018-12-24	Consultation with neurologists	Spinal cord injury, leukemia involvement. Myelitis or medicine-related adverse reaction should all be considered, but the exact cause cannot be identified
2019-01 to 2019-03	BM and CSF examination every month, dexamethasone (10mg, once every 4 day x 4 times) and neurotrophic drugs such as vitamin B12 etc. administered	Continuous CR. The symptoms of spinal cord injury were stable
At the 50 th day after infusion, CAR T cells could not be detected in her PB and BM samples. CAR T cell could not be detected in CSF by quantitative PCR; it contained fewer cells, but CD19+ BC and normal B cells were all negative by FACS		
2019-05-11 (Day 180)	PB and BM examination; CSF examination could not be performed. Head and Spinal MRI	BC in BM more than 20%, Relapse 2 No occupying lesion/mass or other abnormalities found
2019-06-10 (Day 209)	Head CT etc.	Died of leukemia progression (gastrointestinal and intracranial bleeding)

Analysis of the whole treatment protocol for the patient, BC in bone marrow and cerebrospinal fluid were continuously negative within 4 months after CAR T cell treatment indicating that [Redacted] acquired complete remission, no occupying lesions or other abnormalities were found by many times of imaging examinations, no evidence of CNS infection. We think that the spinal cord injury was most likely related to CAR T cell toxicity and defined it as ICANS grade 4. We have added the relevant details to the revised manuscript.

7. Methods: Please add details about the co-stimulatory domain to the manuscript.

We agree with the reviewer's suggestion. As a response, we have added these details to the Supplemental Material text.

This section reads as follows:

CAR design and construction

The construct of Sino19 CAR vector in this study was designed as follows: LTR (retrovirus long terminal repeat), CD19-specific scFv (CD19-scFv), hIgG4-CH2CH3, CD28TM (transmembrane domain), CD137, and CD3 ζ . The single chain (scFv) against human CD19 was derived from anti-CD19 monoclonal antibody FMC63 (Zola, Immunol cell biol. 1991, Dec 69). The scFv codon sequence was optimized and synthesized by GenScript (Nanjing) Co. Ltd and cloned in frame with the human IgG4-CH2CH3 domain, CD28 transmembrane domain and cytoplasmic signaling domain, CD137 cytoplasmic signaling domain, and with the ζ chain of the TCR/CD3 complex in the SFG retroviral backbone. The human IgG4-CH2CH3 domain contains the hinge and CH2CH3 domain derived from 104-327 of UniProtKB/Swiss-Prot P01861.1. CD28 transmembrane domain and cytoplasmic signaling domain was derived from the 69-136 of XP011510499.1. CD137 (4-1BB) cytoplasmic signaling domain was derived from the 214-254 of NCBI reference

sequence NP001552.2. The ζ chain of the TCR/CD3 complex was derived from the 51-163 of NCBI reference sequence NP000725.1. Thus both CD28 and CD137 (4-1BB) costimulatory domains were included in the CAR constructs.

8. Results: What information do you have regarding Tregs and how this was impacted by prior therapies?.

Tregs are decisive not only in the protection against the destruction of own tissues by immunocompetent cells but also in the immunological answer to developing cancers. On the other hand, Tregs could be responsible for the progression of acute and chronic leukemias⁽¹⁻⁴⁾. In the past 2 decades, we also conducted some researches on the relationship between Tregs and tumors, the results were similar to those reported by other scholars⁽⁵⁻⁸⁾.

About the influence of chemotherapy on Tregs, such as lymphodepletion commonly used to promote CAR expansion and survival, depletion of Tregs may be one of the mechanisms, these regimens are nonspecific and provide only a narrow window before Tregs repopulate. Our previous research also showed that Tregs decrease significantly only when tumor cells are effectively removed by chemotherapy, if the tumor recurred or progressed, Tregs level will rise again⁽⁷⁻⁸⁾.

Therefore, we think that elevated Tregs are mainly affected by tumor. Moreover, the levels of Tregs shown in this paper were all detected in patients with refractory/relapsed B-cell acute lymphoblastic leukemia after conditioning chemotherapy and before CAR T cell infusion, results may be slightly affected by the prior therapies.

To our knowledge, the method/medication which can effectively control Tregs are not developed yet, their development would greatly improve the efficacy of CAR T cells. We have been exploring methods which can clear or inhibit Tregs. Recently, we found that a research has made progress in this area⁽⁹⁻¹⁰⁾, which is very impressive.

Related references on this issue are as follows:

- 1) Idris SZ, Hassan N, Lee LJ, et al. Increased regulatory T cells in acute lymphoblastic leukaemia patients. *Hematology*. 2016 May; 21(4): 206-212.
- 2) Takeuchi Y, Nishikawa H. Roles of regulatory T cells in cancer immunity. *Int Immunol*. 2016 Aug; 28(8): 401-409.
- 3) J Duell, M Dittrich, T Bedke, et al. Frequency of regulatory T cells determines the outcome of the T-cell-engaging antibody blinatumomab in patients with B-precursor ALL. *Leukemia*. 2017 Oct; 31(10): 2181–2190.
- 4) M. Niedźwiecki, O. Budziło, E. Adamkiewicz-Drożyńska, et al. CD4⁺CD25^{high}CD127^{low}FoxP3⁺ Regulatory T-Cell Population in Acute Leukemias, A Review of the Literature, *J Immunol Res*. 2019; Published online 2019 Mar 3.
- 5) Zhimin Zhai*, Zimin Sun, Qing Li, et al. Correlation of the CD4⁺CD25^{high} T regulatory cells in recipients and their corresponding donors to acute GVHD, *Transplant International*, 2007; 13 (5) : 400-406
- 6) Li Q, Zhai Zhimin*, Xu X, Shen Y, Zhang A, Sun Z, Liu H, Geng L, Wang Y, Decrease of CD4(+)CD25(+) regulatory T cells and TGF-beta at early immune reconstitution is associated to the onset and severity of graft-versus-host disease following allogeneic haematogenesis stem cell transplantation. *Leukemia Research*, 2010; 34 (9) 1158–1168
- 7) Chen T, Wang H, Zhang Z, Li Q, Yan K, Tao Q, Ye Q, Xiong S, Wang Y, Zhai Z*. A novel cellular senescence gene, SENEX, is involved in peripheral regulatory T cells accumulation in aged urinary bladder cancer. *PLoS One*. 2014 Feb 5; 9(2): e87774. *PLoS One*. 2014; 9(2): e87774.
- 8) Tao Q, Pan Y, Wang Y, Wang H, Xiong S, Li Q, Wang J, Tao L, Wang Z, Wu F, Zhang R, Zhai Z*. Regulatory T cells-derived IL-35 promotes the growth of adult acute myeloid leukemia blasts. *Int J Cancer*. 2015; 137(10):2384-2393.
- 9) Suryadevara CM, Desai R, Farber SH, et al. Preventing Lck Activation in CAR T Cells Confers Treg Resistance but Requires 4-1BB Signaling for Them to Persist and Treat Solid Tumors in Nonlymphodepleted Hosts. *Clin Cancer Res*. 2019; 25(1): 358-368.
- 10) Ohue Y, Nishikawa H. Regulatory T (Treg) cells in cancer: Can Treg cells be a new therapeutic target? *Cancer Sci*. 2019 Jul; 110(7): 2080-2089.

9. Results: it is mentioned that TKIs could be used for Ph+ ALL, but recent data supports that TKI may modulate CAR persistence. What decision making was employed when adding TKI

Yes, some data reported that TKI can modulate CAR persistence, even on/off switch for CAR T cells (Mestermann K, Giavridis T, Weber J, et al. The tyrosine kinase inhibitor dasatinib acts as a pharmacologic on/off switch for CAR T cells, *SciTransl Med.* 2019; 11: 499). But the effect may be temporary and reversible, would not compromise CAR T therapeutic efficacy.

However, in this study, we decided to use TKI only after the loss of CAR T cells persistence in relapsed patients (achieved MRD negativity but lost it to become MRD positive with the acquisition of the Ph+ chromosome). Additionally, this protocol was used for a very limited number of cases. This should have little effect on the CAR T efficacy, but possibly had a little influence to the OS. We would like to analyze bridging other treatments, such as TKI after CR on the efficacy in addition to HSCT, but because this subtype patient too few to do, we will explore this aspect after accumulating more patients in the future.

10. Re: EMD disease, how was this routinely monitored.

The routine protocol for monitoring and evaluating EMDs is as follows:

1) For patients with the EMDs confined to CNS involvement: daily observations of clinical manifestations and detecting various biomarkers in accordance with our research protocol, thus if blast cells in peripheral blood and bone marrow disappear; however, if complete blood count is more than $30 \times 10^9/L$ after CAR T cells infusion or any suspicious symptoms of relapse, then lumbar puncture and CSF examination would be conducted immediately and then every 4-8 weeks until remission is achieved, or every three months in sustained remissions over six months.

2) For patients with EMDs other than CNS involvement: bone marrow and other examinations were conducted regularly, if the blast cells in peripheral blood and bone marrow disappear, re-examination focusing on the lesions detected before CAR T cell infusions were performed by MRI, CT or positron-emission tomography (PET-CT was the first to be considered if possible), then repeated after about 2 months. If the abnormal lesions persist for more than 2 months after CAR T cells infusion, it is regarded as treatment failure, otherwise the patient is monitored regularly every 3 months.

11. Results: did anyone look at the impact of prior blind?

Sorry, we do not quite understand the question. We assumed that the reviewer want to know whether the investigators were blinded to the results of the trial. If so, then the answer is that the investigators were all blinded when they analyzed the results in this study. However, if the question is about treatment blind, then as a response, this does not apply to our trial since it was an open-label study.

Reviewers' comments:

Reviewer #1 (Remarks to the Author):

The revised manuscript addresses most of my critiques pretty well. The manuscript is still very poorly written and will still need to be significantly edited.

Reviewer #2 (Remarks to the Author):

The question that the authors were unable to address was because it was a typographical error. I wanted to know if the authors had looked at the impact of prior blina (blinatumomab) on CD19 CAR response.

In addition, given the varying chemotherapies used, can you look at Treg pattern by conditioning chemotherapy received. Knowing what you know now about the impact of Tregs, did the values correspond with a certain chemotherapy which would make you think about using one over another.

Can you comment if the CNS toxicity/ICANS grade 4 patient developed Guillain Barre. (ascending paralysis)

Reviewer #3 (Remarks to the Author):

The authors evaluated 47 r/r ALL patients received CD19+ CAR-T cell infusion and try to identify prognostic factors, such as EMD and Tregs, to show those are associated with OS and RFS, respectively. The response and safety results look good. Though I have several major concerns.

1. Novelty is not clear

a. Many phase II clinical trials for ALL patients with CD19+ CAR-T have reported their findings, such as Maude et al., 2016, Turtle et al., 2016, etc. The safety and efficacy of this Sino 19 should be compared with those trials and show the difference if any.

b. Some articles were trying to identify factors associated with survival outcomes, see Park et al., 2018, Hay et al., 2019, etc. The association between factors (such as EMD, High-risk cytogenetics, HCT, etc) and EFS (or OS) have already been shown in the univariate setting in Hay 2019. So it's not a novel finding. Some references:

- Park JH, Riviere I, Gonen M, et al. Long-term follow-up of CD19 CAR therapy in acute lymphoblastic leukemia. *N Engl J Med.* 2018; 378(5):449-459.

- Hay, K. A., Gauthier, J., Hirayama, A. V., Voutsinas, J. M., Wu, Q., Li, D., ... & Hawkins, R. M. (2019). Factors associated with durable EFS in adult B-cell ALL patients achieving MRD-negative CR after CD19 CAR T-cell therapy. *Blood, The Journal of the American Society of Hematology*, 133(15), 1652-1663.

c. It's not a novel finding to show correlation of persistence and B Cell Aplasia (figure 5). It's well-known.

2. Recommend obtaining a biostatistician to review data analysis part

a. Kaplan Meier curve is not appropriate to show two group comparison if the group defined by variable happened after day 0. For example, C in both figure 2 and 3 was showing difference between HSCT vs not group. HSCT happened after CAR-T infusion and different patients had different HSCT time. It's inappropriate to use this future information to classify patients' status at baseline (HSCT vs not), and should use time-dependent Cox model instead (delete KM curves). Please find some references as below

- Therneau, T., Crowson, C., & Atkinson, E. (2013). Using time dependent covariates and time dependent coefficients in the cox model. *Red*, 2(1).

- Schultz, L. R., Peterson, E. L., & Breslau, N. (2002). Graphing survival curve estimates for time-dependent covariates. *International journal of methods in psychiatric research*, 11(2), 68-74.

- Bernasconi, D. P., Valsecchi, M. G., & Antolini, L. (2018). Non-parametric estimation of survival probabilities with a time-dependent exposure switch: application to (simulated) heart transplant data. *Epidemiology, Biostatistics and Public Health*, 15(3).

b. It's not clear how the authors conducted multivariable analysis. Did they use some variable selection approach, such as stepwise regression? Did they include more than 1 variables in the model (e.g., EMD, Treg, High-risk cytogenetics, HSCT) and if so HR with 95%CI and p-value from multivariable model should be reported. It seemed that the author only conducted univariate analysis instead of multivariable analysis between EMD and OS, Treg and RFS, respectively. If so, as mentioned as above, those association (univariate) have been reported and not novel. It might have some novelty to report a new multivariable model with multiple factors. Though the sample size is not large, to include 2-3 covariates in the multivariable Cox model should be OK, especially one of the reviewers has pointed that Treg and EMD has strong association (the authors showed this strong association as well in figure 6), those two should be included in a multivariable model together. In addition, just wonder whether the authors have LDH and platelet data? Hay found those two as prognostic factors associated with EFS. If those variables are available, recommend to test as a validation study.

c. Correct typos. For example, it should be 'conduct multivariable analysis' not 'multivariate analyzed'. In table 2, Pearson correlation coefficients were not shown but it was mentioned to be included in statistical section (delete this sentence), etc. Recommend to get a biostatistician to review the stat section/plots/results

Responses and explanations to Reviewer #2 comments:

1. The question that the authors were unable to address was because it was a typographical error. I wanted to know if the authors had looked at the impact of prior blina (blinatumomab) on CD19 CAR response.

I'm sorry that we misunderstood the question at first. About the impact of prior blina (blinatumomab) on CD19 CAR response, total 3 patients had received blinatumomab treatment in our research, the CD19 antigens were all still positive on their leukemia cell surface at relapse and they all achieved remission after CAR T cell infusion. As the reviewer point out, we also worried whether the prior blina (blinatumomab) impact on CD19 CAR response and consulted some documents, but couldn't find any clear answers. So according our result, we think that prior blina (blinatumomab) doesn't impact on CD19 CAR response as long as the CD19 antigen don't loss or still positive at relapse.

2. In addition, given the varying chemotherapies used, can you look at Treg pattern by conditioning chemotherapy received. Knowing what you know now about the impact of Tregs, did the values correspond with a certain chemotherapy which would make you think about using one over another.

Yes, agree the reviewer's thinking. As mentioned before, we indeed want some chemotherapy or other methods can reduce Tregs, but in fact it's very difficult, our previous research showed that Tregs decreased significantly only when tumor cells were effectively removed by chemotherapy, if the tumor recurred or progressed, chemotherapy almost had no influence on Tregs. Tregs were detected after conditioning chemotherapy and before CAR T cell infusion in this study. To further verify and better explain this problem, we analyzed and compared the Tregs in different patient group. The results are follows:

Mann Whitney U test showed no significant difference in Tregs level between the different conditioning chemotherapy groups

	P value
FC vs. VDCP	0.191
FC vs. CTX	0.884
CTX vs. VDCP	0.095

Chi square test also showed no significant difference in the proportion of patients with higher level Tregs between the different conditioning chemotherapy groups (P = 0.097)

	Total No.	Higher Level Treg No.(%)	P value
FC	25	15(60.0%)	0.097
VDCP	5	5(100.0%)	
CTX	14	8(57.1%)	
None	2	0	
Total	46	28(60.9%)	

In addition, we declare that the selection of pre-conditioning chemotherapy in this study wasn't affected by Tregs at all, strictly conducted according to the protocol.

3. Can you comment if the CNS toxicity/ICANS grade 4 patient developed Guillian Barre. (ascending paralysis)

Yes, this patient had similar clinical performances with Guillian-Barre syndrome at starting. We also did suspect this and specifically consulted neurologists to discuss the issue with them. Along with the observation and contemplation over a long time, we all didn't think the patient developed Guillian-Barre syndrome lastly, the reasons as follows:

a) The patient had no history of prodromal infection, and there were no abnormalities in the common relevant

infection tests such as EBV, CMV etc.

b) No protein-cell separation phenomenon in this patient by many times CSF examination.

c) We used gamma globulin, neuronutrition etc., the symptoms of the patient were not relieved. On the contrary, the patient developed persistent bladder and rectal dysfunction later.

By the way, if this patient's condition fully met the diagnostic criteria of Guillian-Barre syndrome, may be also caused by the immune damage associated with CAR T. This is my personal view. I don't know if this is correct.

Responses and explanations to Reviewer #3 comments:

1. Novelty is not clear

a. Many phase II clinical trials for ALL patients with CD19+ CAR-T have reported their findings, such as Maude et al., 2016, Turtle et al., 2016, etc. The safety and efficacy of this Sino 19 should be compared with those trials and show the difference if any.

b. Some articles were trying to identify factors associated with survival outcomes, see Park et al., 2018, Hay et al., 2019, etc. The association between factors (such as EMD, High-risk cytogenetics, HCT, etc) and EFS (or OS) have already been shown in the univariate setting in Hay 2019. So it's not a novel finding. Some references:

- Park JH, Riviere I, Gonen M, et al. Long-term follow-up of CD19 CAR therapy in acute lymphoblastic leukemia. *N Engl J Med.* 2018; 378(5):449-459.

- Hay, K. A., Gauthier, J., Hirayama, A. V., Voutsinas, J. M., Wu, Q., Li, D., ... & Hawkins, R. M. (2019). Factors associated with durable EFS in adult B-cell ALL patients achieving MRD-negative CR after CD19 CAR T-cell therapy. *Blood, The Journal of the American Society of Hematology*, 133(15), 1652-1663.

Response a. and b.:

We agree with the reviewer's comment that the outcome in patients with ALL using CAR-T therapy has been reported by several phases I/II clinical trials, we also carefully reviewed some of these existing data when we summarized and analyzed our results (from references 1 to 12, including several studies and reports recommended by the reviewer: Maude et al., 2016, Turtle et al., 2016, etc.). We found that all the results showed very good remission rates, but they mainly analyzed the influence of CAR T cell on the outcomes, reports on long-term efficacy and other influence factors were limited. Our trial included pediatric and adult patients and most of them had EMDs, so in addition to the safety and efficacy, we mainly analyzed the impact of patient's characteristics on the outcome. We then compared our findings with the results from Maude's et al., because this study had a similar patient population as theirs (2018). Hay et al., 2019, had not published their article at the time the manuscript was developed. We are very grateful to the reviewer for bringing this to our attention. Based on the suggestions, we re-analyzed some results with more appropriate statistical methods and compared them with recent reports. This has been captured in our revised manuscript in the discussion section as well as in Reviewer #2 response. The results showed that EMDs and higher Tregs were independent higher risk factors for death and relapse after CAR T cell therapy by multivariate analysis using the Cox proportional hazard model. Reports like these are limited, especially for Tregs; hence, our research has some novelty.

c. It's not a novel finding to show correlation of persistence and B Cell Aplasia (figure 5). It's well-known.

Response c.

About correlation of persistence and B Cell Aplasia (figure 5), we agree with the reviewers' comments that it's not a novel finding. Currently, the method for detecting B cell in the clinic is unified, very economical and convenient, but it was not been used for CAR T cells in our institution. We considered the potential benefit/usefulness of B cell aplasia as a surrogate marker for predicting CAR T cell survival in vivo if it could be recognized and accepted, so we analyzed and presented the result specifically for that purpose. We also assumed that it may benefit other institutions. However, if the

reviewer thinks it is not necessary, we can delete it.

2. Recommend obtaining a biostatistician to review data analysis part

a. Kaplan Meier curve is not appropriate to show two group comparison if the group defined by variable happened after day 0. For example, C in both figure 2 and 3 was showing difference between HSCT vs not group. HSCT happened after CAR-T infusion and different patients had different HSCT time. It's inappropriate to use this future information to classify patients' status at baseline (HSCT vs not), and should use time-dependent Cox model instead (delete KM curves). Please find some references as below

- Therneau, T., Crowson, C., & Atkinson, E. (2013). Using time dependent covariates and time dependent coefficients in the cox model. *Red*, 2(1).
- Schultz, L. R., Peterson, E. L., & Breslau, N. (2002). Graphing survival curve estimates for time - dependent covariates. *International journal of methods in psychiatric research*, 11(2), 68-74.
- Bernasconi, D. P., Valsecchi, M. G., & Antolini, L. (2018). Non-parametric estimation of survival probabilities with a time-dependent exposure switch: application to (simulated) heart transplant data. *Epidemiology, Biostatistics and Public Health*, 15(3).

We are very grateful for the reviewer's constructive and professional comments on data analysis and statistical processing. We have used the recommended analysis methods and revised our manuscript accordingly under the results section. Time-dependent Cox model has been used for the comparison between HSCT vs. not group, KM curves of figure 2C and 3C in the previous manuscript have been deleted. The details as follows:

Step1. We added a new variable: waiting time for HSCT- "waittime".

'414/41/35/.../41' are the actual days of waiting for HSCT

'99999' represents infinite waiting, means not receiving a HSCT

RFS	Relapse	Bridging HSCT	waittime
712	1	1	414
319	1	1	41
718	0	1	35
704	0	1	143
481	1	1	55
475	1	1	40
312	0	1	55
194	0	1	149
163	0	1	88
86	0	1	41
37	1	0	999999
36	1	0	999999
276	1	0	999999
95	1	0	999999
1089	0	0	999999
207	1	0	000000

Step2. Establishing the time-dependent covariate 'T_COV_'

The expression is as follows: $(T_{<waittime|waittime=999999}) * 0 + (T_{>waittime}) * 1$

Step3. We included both 'T_COV_' and 'BridgingHSCT', used the time-dependent Cox model with the method of 'forward stepwise (likelihood ratio)' to analyze the effect of HSCT on survival, the results are as follows:

i. HSCT on OS:

Block 1: Method = Forward Stepwise (Likelihood Ratio)

Omnibus Tests of Model Coefficients^b

Step	-2 Log Likelihood	Overall (score)			Change From Previous Step			Change From Previous Block		
		Chi-square	df	Sig.	Chi-square	df	Sig.	Chi-square	df	Sig.
1 ^a	94.364	4.334	1	.037	5.677	1	.017	5.677	1	.017

a. Variable(s) Entered at Step Number 1: BridgingHSCT
 b. Beginning Block Number 1. Method = Forward Stepwise (Likelihood Ratio)

Variables in the Equation

	B	SE	Wald	df	Sig.	Exp(B)	95.0% CI for Exp(B)	
							Lower	Upper
Step 1 BridgingHSCT	-1.871	1.034	3.275	1	.070	.154	.020	1.168

Variables not in the Equation^a

	Score	df	Sig.
Step 1 T_COV_	.155	1	.694

a. Residual Chi Square = .155 with 1 df Sig. = .694

Model if Term Removed

Term Removed	Loss Chi-square	df	Sig.
Step 1 BridgingHSCT	5.677	1	.017

Covariate Means

	Mean
T_COV_	.249
BridgingHSCT	.301

Results: 'T_COV_' was excluded from Equation (P=0.694) and indicates that 'waittime' did not affect the OS. HSCT was a protective factor for OS (P=0.070) and patients who bridged to a HSCT had a lower risk of death (HR=0.154, 95%CI =0.020~1.168).

ii. HSCT on RFS

Block 1: Method = Forward Stepwise (Likelihood Ratio)

Omnibus Tests of Model Coefficients^b

Step	-2 Log Likelihood	Overall (score)			Change From Previous Step			Change From Previous Block		
		Chi-square	df	Sig.	Chi-square	df	Sig.	Chi-square	df	Sig.
1 ^a	123.915	4.598	1	.032	5.107	1	.024	5.107	1	.024

a. Variable(s) Entered at Step Number 1: BridgingHSCT
 b. Beginning Block Number 1. Method = Forward Stepwise (Likelihood Ratio)

Variables in the Equation

	B	SE	Wald	df	Sig.	Exp(B)	95.0% CI for Exp(B)	
							Lower	Upper
Step 1 BridgingHSCT	-1.142	.560	4.164	1	.041	.319	.107	.956

Variables not in the Equation^a

	Score	df	Sig.
Step 1 T_COV_	1.770	1	.183

a. Residual Chi Square = 1.770 with 1 df Sig. = .183

Model if Term Removed

Term Removed	Loss Chi-square	df	Sig.
Step 1 BridgingHSCT	5.107	1	.024

Covariate Means

	Mean
T_COV_	.194
BridgingHSCT	.364

Results: 'T_COV_' was also excluded from Equation (P=0.183) and indicates that 'waittime' did not affect the RFS. HSCT was a protective factor for RFS (P=0.041) and patients who bridged a HSCT had a lower risk of relapse (HR=0.319, 95%CI 0.107-0.956).

Lastly, the results analyzed by the time-dependent Cox model were consistent with that reported by Hay et al.

b. It's not clear how the authors conducted multivariable analysis. Did they use some variable selection approach, such as stepwise regression? Did they include more than 1 variables in the model (e.g., EMD, Treg, High-risk cytogenetics, HSCT) and if so HR with 95%CI and p-value from multivariable model should be reported. It seemed that the author only conducted univariate analysis instead of multivariable analysis between EMD and OS, Treg and RFS, respectively. If so, as mentioned as above, those association (univariate) have been reported and not novel. It might have some novelty to report a new multivariable model with multiple factors. Though the sample size is not large, to include 2-3 covariates in the multivariable Cox model should be OK, especially one of the reviewers has pointed that Treg and EMD has strong association (the authors showed this strong association as well in figure 6), those two should be included in a multivariable model together. In addition, just wonder whether the authors have LDH and platelet data? Hay found those two as prognostic factors associated with EFS. If those variables are available, recommend to test as a validation study.

We thank the reviewer for this positive assessment and agree with the suggestions made. We are very sorry for the process of multivariable analysis didn't display clearly. In response, the manuscript has been revised.

About variable selection approach, yes, we used 'Forward Stepwise (Likelihood Ratio)' which is stepwise regression based on maximum likelihood estimation, but we did not clarify it well in the previous manuscript. The prior Kaplan

Meier analysis showed that EMDs, Tregs, and High-risk cytogenetics were associated with both RFS/OS. We mainly focused on the influence of patient characteristics on CAR T unitary therapy, so we just included the three variables (EMDs, Tregs, and High-risk cytogenetics) in the multivariate analysis using Cox proportional hazard model, whilst HSCT was excluded as an additional treatment. The details as follows:

i. The output result of multivariable analysis on OS:

Block 1: Method = Forward Stepwise (Likelihood Ratio)

Omnibus Tests of Model Coefficients^b

Step	-2 Log Likelihood	Overall (score)			Change From Previous Step			Change From Previous Block		
		Chi-square	df	Sig.	Chi-square	df	Sig.	Chi-square	df	Sig.
1 ^a	90.008	9.604	1	.002	7.763	1	.005	7.763	1	.005

a. Variable(s) Entered at Step Number 1: EMD
b. Beginning Block Number 1. Method = Forward Stepwise (Likelihood Ratio)

Variables in the Equation

	B	SE	Wald	df	Sig.	Exp(B)	95.0% CI for Exp(B)	
							Lower	Upper
Step 1 EMD	1.515	.532	8.120	1	.004	4.551	1.605	12.907

Variables not in the Equation^a

	Score	df	Sig.
Step 1 HighriskCytogenetic	2.628	1	.105
HighTreg	1.140	1	.286

a. Residual Chi Square = 3.518 with 2 df Sig. = .172

Model if Term Removed

Term Removed	Loss Chi-square	df	Sig.
Step 1 EMD	7.763	1	.005

EMD: HR =4.551, 95%CI 1.605~12.907, P=0.004

Because the Forward Stepwise method was adopted, high-risk cytogenetic and Tregs were not included in the Equation, we did not get the HR and P values for these two meaningless variables from the SPSS software output.

ii. The output result of multivariable analysis on RFS:

Block 1: Method = Forward Stepwise (Likelihood Ratio)

Omnibus Tests of Model Coefficients^b

Step	-2 Log Likelihood	Overall (score)			Change From Previous Step			Change From Previous Block		
		Chi-square	df	Sig.	Chi-square	df	Sig.	Chi-square	df	Sig.
1 ^a	63.889	7.829	1	.005	7.383	1	.007	7.383	1	.007

a. Variable(s) Entered at Step Number 1: HighTreg
 b. Beginning Block Number 1. Method = Forward Stepwise (Likelihood Ratio)

Variables in the Equation

	B	SE	Wald	df	Sig.	Exp(B)	95.0% CI for Exp(B)	
							Lower	Upper
Step 1 HighTreg	1.628	.631	6.656	1	.010	5.096	1.479	17.558

Variables not in the Equation^a

	Score	df	Sig.
Step 1 HighriskCytogenetic	2.164	1	.141
EMD	1.782	1	.182

a. Residual Chi Square = 3.265 with 2 df Sig. = .195

Model if Term Removed

Term Removed	Loss Chi-square	df	Sig.
Step 1 HighTreg	7.383	1	.007

HighTreg: HR=5.096, 95%CI 1.479~17.558, P=0.010.

Because the Forward Stepwise method was adopted, high-risk cytogenetic and EMD were not included in the Equation, we also did not get the HR and P values for these two meaningless variables from the SPSS software output.

With regards to the reviewer’s recommendation on the inclusion of LDH and platelet as variables, we agree and we are also very interested. Higher LDH and lower platelet levels are two general indicators of poor prognosis in hematologic tumors, but can also easily be influenced by various factors such as chemotherapy and infection, etc., so we worry that it is difficult to ensure the consistency and comparability of these two values for each patient if now add these two variables. Secondly, just as Hay et al. (2019) analyzed and discussed in their paper, “high LDH concentration and low platelet count correlated with the need for bridging systemic therapy. Serum LDH concentration has been associated with tumor burden and proliferative activity in B-cell and other malignancies and may correlate with an immunosuppressive tumor microenvironment. Thrombocytopenia could also be due to the cumulative effects of multiple previous therapy regimens, which in turn could be associated with poor T cell function”, so from the perspective of patient factors, we think these results are consistent with our findings. We very thank for the reviewer’s suggestion, our clinical trial is still ongoing, and we hope to incorporate these two variables in our next publication.

c. Correct typos. For example, it should be ‘conduct multivariable analysis’ not ‘multivariate analyzed’. In table 2, Pearson correlation coefficients were not shown but it was mentioned to be included in statistical section (delete this sentence), etc. Recommend to get a biostatistician to review the stat section/plots/results

We thank the reviewer for pointing out the typo mistakes. We have corrected them in the revised manuscript and deleted the phrase ‘Pearson correlation coefficients’. In addition, we invited Xu Zhang (Ph.D.) from the Department of Epidemiology and Biostatistics, School of Public Health, Anhui Medical University to check our statistical analyses again.

REVIEWER COMMENTS

Reviewer #2 (Remarks to the Author):

No additional comments. The authors have taken the necessary steps to address all reviewer comments and provide detail in their approach.

Reviewer #3 (Remarks to the Author):

The authors tried to address my stat concerns and it's good that they have a biostatistician to review the analysis and results. Though some of their update analysis is not correct. See my comments as below:

1. Time-dependent Cox model: the author used software SPSS to conduct the analysis, and I am not familiar with this software. Based on the online tutorial, https://www.ibm.com/support/knowledgecenter/fi/SSLVMB_24.0.0/spss/advanced/idh_coxt.html, the author generated a "segmented time-dependent covariate" by step 1 and step 2, which is good. But step 3 is not correct. Since the authors has generated "T_COV_", they don't need to include both "T_COV_" and "Bridging HSCT" in the model and they don't need to use method "forward stepwise". Just keep "T_COV_" in and run "Cox time-dependent model" for OS and RFS, respectively.

Please search "Cox proportional hazard regression with time varying covariate in spss" (a youtube video by Ayumi Shintani) and it will show how to do time-dependent Cox model step by step. If possible, recommend using other stat software, such as R to run the analysis and attached source code online.

2. It's ok to use forward stepwise regression and the results by selecting EMD for OS and Treg for RFS is acceptable. Though stepwise regression is well known to poorly pick key factors if several covariates are highly correlated. Suggest to add one multivariable model using all three variables in (EMD, Treg, and cytogenetics) to show HR, CI and p-value. Though some might not be significant, HR will provide informative information on the direction.

3. There are still some typos, e.g., please change "multivariate" to be "multivariable".

Responses to Reviewer #3 comments:

1. Time-dependent Cox model: the author used software SPSS to conduct the analysis, and I am not familiar with this software. Based on the online tutorial, https://www.ibm.com/support/knowledgecenter/fi/SSLVMB_24.0.0/spss/advanced/idh_coxt.html, the author generated a “segmented time-dependent covariate” by step 1 and step 2, which is good. But step 3 is not correct. Since the authors has generated “T_COV_”, they don’t need to include both “T_COV_” and “Bridging HSCT” in the model and they don’t need to use method “forward stepwise”. Just keep “T_COV_” in and run “Cox time-dependent model” for OS and RFS, respectively.

Please search “Cox proportional hazard regression with time varying covariate in spss” (a youtube video by Ayumi Shintani) and it will show how to do time-dependent Cox model step by step.

If possible, recommend using other stat software, such as R to run the analysis and attached source code online.

We are grateful to the reviewer for bringing this to our attention; we appreciate the helpful comments and suggestions. We searched out the youtube video by Ayumi Shintani and studied carefully “Cox proportional hazard regression with time-varying covariate in spss” over and over. We agree with the reviewer, we had some misunderstandings about Step 3. Based on the video’s tutorial, we re-analyzed the data and corrected the mistake in Step 3 (we just kept “T_COV_” in and re-run the “Cox time-dependent model” for OS and RFS). The results were as follows:

1) HSCT on OS:

Block 1: Method = Enter

Omnibus Tests of Model Coefficients^a

-2 Log Likelihood	Overall (score)			Change From Previous Step			Change From Previous Block		
	Chi-square	df	Sig.	Chi-square	df	Sig.	Chi-square	df	Sig.
95.737	3.290	1	.070	4.304	1	.038	4.304	1	.038

a. Beginning Block Number 1. Method = Enter

Variables in the Equation

	B	SE	Wald	df	Sig.	Exp(B)	95.0% CI for Exp(B)	
							Lower	Upper
T_COV_	-1.678	1.035	2.628	1	.105	.187	.025	1.420

2) HSCT on RFS:

Block 1: Method = Enter

Omnibus Tests of Model Coefficients^a

-2 Log Likelihood	Overall (score)			Change From Previous Step			Change From Previous Block		
	Chi-square	df	Sig.	Chi-square	df	Sig.	Chi-square	df	Sig.
127.767	1.176	1	.278	1.255	1	.263	1.255	1	.263

a. Beginning Block Number 1. Method = Enter

Variables in the Equation

	B	SE	Wald	df	Sig.	Exp(B)	95.0% CI for Exp(B)	
							Lower	Upper
T_COV_	-.630	.589	1.145	1	.285	.533	.168	1.689

The outputs showed that HR for HSCT on OS and RFS was 0.187 (95%CI 0.025~1.420) and 0.533(95%CI: 0.168~1.689) respectively, still indicating that HSCT is a protective factor for OS and RFS, though not statistically significant (P=0.105 for OS and P=0.285 for RFS). Reference to a recent report by Hay et al. demonstrated that patients undergoing alloHSCT after CAR-T cell therapy had a lower risk of failure for EFS compared to those who did not undergo HSCT in a univariable and multivariable stepwise modeling analysis. On the contrary, Park et al. found no significant difference in EFS and OS between patients who underwent transplantation and those who did not by the log-rank test. Analyzing these results comprehensively, we think whether patients can derive benefit from alloHSCT after CAR-T cell therapy need further definitive randomized studies to validate, but as a standard treatment for R/R B-ALL and for patients who are eligible for transplantation, alloHSCT should be considered. We have corrected the results and added these explanations to the revised manuscript.

We were unable to use the R stat software to run the analysis as recommended by the reviewer, we are not familiar with it and because SPSS is widely used in our settings, we still used the SPSS to run the analysis. We are sorry about this. However, we will gradually learn how to use R software for our future articles.

2. It's ok to use forward stepwise regression and the results by selecting EMD for OS and Treg for RFS is acceptable. Though stepwise regression is well known to poorly pick key factors if several covariates are highly correlated. Suggest to add one multivariable model using all three variables in (EMD, Treg, and cytogenetics) to show HR, CI and p-value. Though some might not be significant, HR will provide informative information on the direction.

We appreciated the reviewer's comments. In agreement with the reviewer's suggestion, we considered and factored it into our initial analysis. We used two analysis methods when we conducted the Cox proportional hazard multivariable model to identify factors associated with OS and RFS: one used the method 'Forward Stepwise (Likelihood Ratio)', which was presented in our manuscript; the other one used method 'Enter', which included all three variables. The outputs by the latter method showed that the three variables (EMD, Treg and cytogenetics) were all poor factors for OS and RFS, EMD had the highest HR for OS (HR=2.792) and Treg had the highest HR for RFS (HR=3.104), but all of them were not statistically significant. We had consulted several biostatisticians, they all thought that both methods were reasonable and acceptable, hence we settled on the stepwise regression method in our manuscript. The stepwise regression method was used in several similar types of research, and the statistically significant factors got from it were considered to be independent or associated, including the article recommended by the reviewer. (Hay, K. A., Gauthier, J., Hirayama, A. V., Voutsinas, J. M., Wu, Q., Li, D., ... & Hawkins, R. M. Factors associated with durable EFS in adult B-cell ALL patients achieving MRD-negative CR after CD19 CAR T-cell therapy. *Blood, The Journal of the American Society of Hematology*, 2019; 133(15), 1652-1663). We side with the reviewer that HR will provide information on the direction of relative risk, even if not significant. Upon comparing the results in these two models, the information and direction expressed by both models were consistent, with a slight difference in significance. Considering the above factors, we still selected the stepwise regression method finally and kept these results in the manuscript.

The results (HR, CI and p-value) by selection method 'Enter' (included all three variables EMD, Treg, and cytogenetics in the Equation) are as follows:

1) Factors associated with OS

Block 1: Method = Enter

Omnibus Tests of Model Coefficients^a

-2 Log Likelihood	Overall (score)			Change From Previous Step			Change From Previous Block		
	Chi-square	df	Sig.	Chi-square	df	Sig.	Chi-square	df	Sig.
86.302	12.521	3	.006	11.469	3	.009	11.469	3	.009

a. Beginning Block Number 1. Method = Enter

Variables in the Equation

	B	SE	Wald	df	Sig.	Exp(B)	95.0% CI for Exp(B)	
							Lower	Upper
HighriskCytogenetic	.968	.647	2.237	1	.135	2.634	.740	9.369
EMD	1.027	.653	2.474	1	.116	2.792	.777	10.033
HighTreg	.587	.651	.813	1	.367	1.798	.502	6.435

	HR	95%CI	P value
Cytogenetics	2.634	0.740~9.369	0.135
EMD	2.792	0.777~10.033	0.116
Treg	1.798	0.502~6.435	0.367

2) Factors associated with RFS

Block 1: Method = Enter

Omnibus Tests of Model Coefficients^a

-2 Log Likelihood	Overall (score)			Change From Previous Step			Change From Previous Block		
	Chi-square	df	Sig.	Chi-square	df	Sig.	Chi-square	df	Sig.
60.445	12.022	3	.007	10.827	3	.013	10.827	3	.013

a. Beginning Block Number 1. Method = Enter

Variables in the Equation

	B	SE	Wald	df	Sig.	Exp(B)	95.0% CI for Exp(B)	
							Lower	Upper
HighriskCytogenetic	1.056	.838	1.589	1	.208	2.875	.557	14.848
EMD	.732	.716	1.047	1	.306	2.080	.511	8.461
HighTreg	1.133	.698	2.633	1	.105	3.104	.790	12.191

	HR	95%CI	P value
Cytogenetics	2.875	0.557~14.848	0.208
EMD	2.080	0.511~8.461	0.306
Treg	3.104	0.790~12.191	0.105

3. There are still some typos, e.g., please change “multivariate” to be “multivariable”.

Thank you for pointing out the typos. We carefully checked the manuscript and corrected the mistakes.

Reviewers' comments:

Reviewer #3 (Remarks to the Author):

The authors addressed my concerns. Looks good!